



# A fully coupled solid particle microphysics scheme for stratospheric aerosol injections within the aerosol-chemistry-climate-model SOCOL-AERv2

Sandro Vattioni[1,2], Rahel Weber[1,2], Aryeh Feinberg[1,3,5], Andrea Stenke[1,3,4], John A. Dykema[2], Beiping Luo[1], Georgios A. Kelesidis[6,7], Christian A. Bruun[6], Timofei Sukhodolov[8], Frank N. Keutsch[2,9,10], Thomas Peter[1], and Gabriel Chiodo[1]

[1]Institute for Atmospheric and Climate Science, ETH Zurich, Switzerland
[2]John A. Paulson School of Engineering and Applied Sciences, Harvard University, Cambridge, MA, USA
[3]Institute of Biogeochemistry and Pollutant Dynamics, ETH Zurich, Switzerland
[4]Eawag, Swiss Federal Institute of Aquatic Science and Technology, Dübendorf, Switzerland
[5]now at: Department of Atmospheric Chemistry and Climate, Institute of Physical Chemistry Blas Cabrera, CSIC, Madrid, Spain
[6]Particle Technology Laboratory, ETH Zurich, Switzerland
[7]now at: Environmental and Occupation Health Science Institute, School of Public Health, Rutgers, The State University of New Jersey, Piscataway, NJ, USA
[8]Physikalisch-Meteorologisches Observatorium Davos and World Radiation Center, Davos, Switzerland
[9]Department of Chemistry and Chemical Biology, Harvard University, Cambridge, MA, USA
[10]Department of Earth and Planetary Sciences, Harvard University, Cambridge, MA, USA

**Correspondence:** Sandro Vattioni (sandro.vattioni@env.ethz.ch)

**Abstract.** Recent studies have suggested that injection of solid particles such as alumina and calcite particles for stratospheric aerosol injection (SAI) instead of sulfur-based injections could reduce some of the adverse side effects of SAI such as ozone depletion and stratospheric heating. Here, we present a version of the global aerosol-chemistry-climate model SOCOL-AERv2 and the Earth System Model (ESM) SOCOLv4 which incorporate a solid particle microphysics scheme for assessment of SAI of solid particles. Microphysical interactions of the solid particle with the stratospheric sulfur cycle were interactively coupled to the heterogeneous chemistry scheme and the radiative transfer code (RTC) for the first time within an ESM. Therefore, the model allows simulation of heterogeneous chemistry at the particle surface as well as feedbacks between microphysics, chemistry, radiation and climate. We show that sulfur-based SAI results in a doubling of the stratospheric aerosol burden compared to the same injection rate of calcite and alumina particles with radius of 240 nm, mainly due to the smaller density and the smaller average particle size of sulfuric acid aerosols and thus, slower sedimentation. Therefore, to achieve the same radiative forcing, larger injection rates are needed for calcite and alumina particle injection than for sulfur-based SAI. The stratospheric sulfur cycle would be significantly perturbed, with a reduction in stratospheric sulfuric acid burden by 53%, when injecting 5 Mt/yr of alumina or calcite particles of 240 nm radius. We show that alumina particles will acquire a sulfuric acid coating equivalent of about 10 nm thickness, if the sulfuric acid is equally distributed over the whole available particle surface area in the lower stratosphere. However, due to a steep contact angle of sulfuric acid on alumina particles, the sulfuric acid coating would likely not cover the entire alumina surface, which would result in available surface for heterogeneous reactions





other than the ones on sulfuric acid. When applying realistic uptake coefficients of 1.0, $10^{-5}$ and $10^{-4}$ for $H_2SO_4$, HCl and $HNO_3$, respectively, the same scenario with injections of calcite particles results in 94% of the particle mass remaining in the form of $CaCO_3$. This likely keeps the optical properties of the calcite particles intact, but could significantly alter the

heterogeneous reactions occurring on the particle surfaces. The major process uncertainties of solid particle SAI are 1) the solid particle microphysics in the injection plume and degree of agglomeration of solid particles on the sub-ESM grid scale, 2) the scattering properties of the resulting agglomerates 3) heterogeneous chemistry on the particle surface and 4) aerosol-cloud interactions. These uncertainties can only be addressed with extensive, coordinated, experimental and modelling research efforts. The model presented in this work offers a useful tool for sensitivity studies and impact analysis of new experimental

results on points 1) to 3) for SAI of solid particles.

## 1  Introduction

Even if greenhouse gas (GHG) emissions stopped today, high GHG concentrations and their effects would persist for centuries, if GHG removal techniques can not be scaled up fast enough (IPCC, 2023). Stratospheric aerosol injection (SAI) has the potential to rapidly mitigate some of the adverse impacts of climate warming by increasing the Earth's albedo. SAI would

be feasible at relatively low cost (Smith, 2020, i.e., about $ 18 billion per year), but it also entails considerable risks such as adverse environmental side effects and challenges such as governance of ethical considerations on global and inter-generational equity and the power of decision (e.g. Robock, 2008; Burns, 2011). For these reasons, the US National Academy of Sciences and others have proposed research, which explores the risks risks and benefits of SAI (e.g., Shepherd, 2012; Schäfer et al., 2015; National Research Council, 2015; Field et al., 2021).

The idea of SAI evolved from the temporary cooling effect of sulfuric acid aerosols formed after stratospheric $SO_2$ injections of large explosive volcanic eruptions and was first proposed by Budyko (1974). In addition, solid particles as alternative materials were explored in conceptual studies and reports on various climate engineering techniques (e.g. Keith and Dowlatabadi, 1992; Teller et al., 1996; Keith, 2000). However, research on SAI was initially a taboo among researchers since it does not present an actual solution to climate change, but instead at best a treatment of some of its symptoms (MacMartin et al., 2014;

Keith and Macmartin, 2015). The need for research on SAI only became more to the forefront with the growing appearance of impacts of climate change and after the proposal to investigate the risks, benefits and the feasibility of SAI by Crutzen (2006). Potential scenarios for SAI involve reducing the current rate of climate change or in what is referred to as an "overshoot scenario", where SAI would aim at keeping global temperature increase below 1.5 K, the target set by the Paris agreement in 2015, until global net zero GHG emissions are achieved, and until solutions are found on how to remove GHG from the air efficiently

(MacMartin et al., 2014; Keith and Macmartin, 2015; MacMartin et al., 2022).

To date, research on SAI has mainly focused on injection of sulfuric acid aerosol precursor species such as $SO_2$. This has a number of reasons: Due to the natural occurrence of sulfuric acid aerosols in the atmosphere, the stratospheric sulfur cycle is relatively well known and interactively simulated in many chemistry climate models (e.g. Thomason and Peter, 2006; Deshler, 2008; Feinberg et al., 2019; Brodowsky et al., 2023), which makes it easier for modellers to investigate sulfur-based SAI



scenarios. Furthermore, heterogeneous chemistry and optical properties of sulfuric acid aerosols in the stratosphere are also relatively well known from experimental studies (e.g., Burkholder et al., 2020; Ammann et al., 2013). In addition, observations after large explosive volcanic eruptions such as the Mt. Pinatubo eruption 1991 (e.g., Arfeuille et al., 2013; Thomason et al., 2018; Baran and Foot, 1994; Kovilakam et al., 2020) allow for model evaluation of the chemical and radiative impacts of large stratospheric sulfur emissions (e.g., Deshler et al., 2019; Quaglia et al., 2023).

However, SAI via $SO_2$ injections also are subject to several limitations making sulfuric acid aerosols less attractive for a potential use in SAI. These limitations include (1) aerosol size distributions that are inefficient for backscattering solar radiation with either too many large or too many small particles (Vattioni et al., 2019), (2) ozone depletion due to chlorine activation on aerosols (Tilmes et al., 2008; Weisenstein et al., 2022), (3) stratospheric warming resulting in changes of the large-scale atmospheric circulation (Aquila et al., 2014; Tilmes et al., 2017; Visioni et al., 2021; Jones et al., 2022; Wunderlin et al., 2024)
as well as (4) substantial inter-model uncertainties on resulting stratospheric aerosol burden and radiative effects (Weisenstein et al., 2022).

Recent studies have shown that injection of solid particles could overcome several of these limitations (e.g., Pope et al., 2012; Weisenstein et al., 2015; Keith et al., 2016; Dykema et al., 2016). Many solid particle candidates such as alumina ($Al_2O_3$), calcite ($CaCO_3$) or diamond particles have larger backscatter efficiencies per stratospheric burden compared to sulfuric
acid aerosols (Dykema et al., 2016). Furthermore, the absorption efficiency of longwave (LW) and shortwave (SW) radiation per resulting aerosol burden is significantly smaller for many solid materials compared to sulfuric acid aerosols, resulting in reduced stratospheric warming. Other studies showed that the injection of alumina or calcite particles would result in less ozone depletion (Weisenstein et al., 2015; Dai et al., 2020) or even in ozone increase in the case of calcite particles (Keith et al., 2016).

However, contrary to sulfuric acid aerosols, the solid particle types proposed for SAI do not occur naturally in the stratosphere. Therefore, relatively little is known about their microphysical interactions and chemical ageing processes, which could alter their scattering properties, their stratospheric residence time as well as heterogeneous chemistry hosted on the particles. This makes it very difficult to have confidence in the modeled impacts of solid particle injections on stratospheric chemistry and climate.

There have been investigations on the impact of alumina-containing solid-fuel space rocket exhaust on stratospheric ozone and radiative forcing. These studies used flow-tube experiments (Molina et al., 1997), 2D-chemistry transport modelling (Jackman et al., 1998; Danilin et al., 2001) as well as conceptual methods (Ross and Sheaffer, 2014). However, the rocket exhaust investigated in these studies also contains other species such as water, HCl and black carbon, which makes attribution of the alumina particles effects on ozone alteration and radiative forcing difficult (Vattioni et al., 2023b). Therefore, microphyiscal in-
teractions of solid particles with background aerosols, as well as their impact on stratospheric chemistry and radiative forcing, remain subject to large uncertainties.

Nevertheless, there have been several studies that investigated SAI scenarios using solid particles. Fujii (2011) and Pope et al. (2012) were among the first conceptual studies which pointed at potential benefits, such as better scattering properties, form SAI of various solid materials in their studies. At the same time Ferraro et al. (2011) and Ferraro et al. (2015) used an





RTC and a general circulation model, respectively, to quantify stratospheric heating resulting from some materials as well as the dynamical stratospheric feedbacks, while prescribing the stratospheric solid particle number densities. Later, Jones et al. (2016) was the first study that compared tropospheric climate impacts from SAI of sulfuric acid aerosols with injections of $TiO_2$ and BC using a global circulation model with an interactive ocean module, while simulating injection and transport of solid particles with prescribed size distributions. However, non of these studies accounted for heterogeneous chemistry on particle

surfaces nor for microphysical processes. Impacts on stratospheric ozone from SAI of solid particles were first assessed by Kravitz et al. (2012) who investigated SAI with BC aerosols using a chemistry climate model. In summary, the conclusion from these first studies which mainly investigated SAI of BC and $TiO_2$ particles is that these materials are not suitable as injection species for SAI since both, $TiO_2$ and BC have strong UV-VIS absorption, which results in significant stratospheric heating. However, while injection of BC would result in substantial ozone depletion, experimental studies on heterogeneous

chemistry on $TiO_2$ surfaces indicated reduced impacts on modelled stratospheric ozone (Tang et al., 2014, 2016; Moon et al., 2018) compared to sulfuric acid aerosols, providing additional motivation for exploration of other species.

Meanwhile, Dykema et al. (2016) performed detailed radiative transfer calculations of various solid particles, including feedbacks from stratospheric water vapor and concluded that solid particles such as calcite, diamond, alumina and SiC scatter solar radiation with better mass efficiency and less stratospheric heating compared to sulfuric acid aerosols. Weisenstein et al. (2015)

was the first study to use a 2D chemistry transport model with interactive solid particle microphysics as well as microphysical interactions of solid particles with condensed and gaseous sulfuric acid to assess impacts from heterogeneous chemistry on alumina particle surfaces. The resulting zonal mean number concentrations were then fed into a RTC offline to simulate the resulting radiative forcing. Limitations of this study stem from a simplified representation of heterogeneous chemistry on alumina particles (Vattioni et al., 2023b) as well as from the 2D approach which causes significant simplifications in atmospheric

dynamics and transport of the injected particles. Keith et al. (2016) used the same model to propose substantial stratospheric ozone increase through removal of HCl from the stratosphere via uptake on calcite particle surfaces and subsequent sedimentation. Later, Cziczo et al. (2019) pointed to the over simplified assessment used in the latter study, which applied over simplified heterogeneous chemistry such as neglecting the formation of hydrates as well as a potential sealing effect due to the formation of reaction products at the surface. However, most importantly, this latter study showed that especially $CaCO_3$ and $Ca(NO_3)_2$

as well as their hydrates are good ice nucleation materials, which could result in in a 33% reduction of the radiative forcing compared to Keith et al. (2016) due to increased cirrus cloud coverage. Furthermore, the interactions of aerosols with polar stratospheric clouds could create a feedback on polar ozone concentrations, which has not been investigated so far (Cziczo et al., 2019).

Therefore, to assess the real risks and benefits of SAI of solid particles compared to the more conventionally researched sulfur

based approach, it is important to interactively couple 1) microphysical processes such as agglomeration and sedimentation of solid particles and their microphysical interaction with condensed and gaseous sulfuric acid with 2) heterogeneous chemistry on the particle surface and the subsequent impacts on stratospheric ozone and with 3) interactive aerosol cloud interactions, as well as with 4) the resulting dynamical feedbacks from changes in ozone, stratospheric warming and cooling of tropospheric climate interactively in one model. Simulating all these effects in a self-consistent way is crucial, because (1) strong agglomeration





can significantly decrease the backscatter efficiency or increase the sedimentation speed compared to a compact monomer, while (2) can lead to significant ozone alteration depending on the material and (3) can result in a positive or negative feedback on radiative forcing through cirrus cloud alteration (e.g., Cziczo et al., 2019). The combination of these processes ultimately determines the large-scale circulation response and surface climate response to SAI with alternate materials.

This study presents a microphysics module for solid particles within the aerosol-chemistry-climate model SOCOL-AERv2,
which represents injected solid particles interactively coupled to advection and sedimentation as well as to the model's radiative transfer and heterogeneous chemistry modules (see Figure 1). Additionally, the module calculates microphysical interactions between solid particles and background sulfuric acid in gaseous and condensed form online. This allows us to account for feedbacks between different processes, which enables to comprehensively assess the risks and benefits of SAI of solid particles. However, it has to be kept in mind that direct aerosol-cloud interactions are not considered in this model, which could alter the
resulting radiative forcing through cirrus cloud feedbacks (e.g., Cziczo et al., 2019). In this study, we focus on the injection of alumina and calcite particles since these are some of the few potential particle types for which some heterogeneous reaction rates have previously been measured (Molina et al., 1997; Huynh and McNeill, 2020; Dai et al., 2020; Huynh and McNeill, 2021).

## 2 Model description

The interactive coupling of aerosol microphysics with heterogeneous chemistry and radiation makes the SOCOL models (Feinberg et al., 2019; Sukhodolov et al., 2021) suitable to explore feedbacks between microphysics, stratospheric chemistry, radiation as well as tropospheric and stratospheric climate. The SOCOL model family has been successfully used to reproduce the global sulfur cycle under volcanically active (e.g., Mt. Pinatubo 1991) and quiescent conditions (e.g., Sheng et al., 2015; Sukhodolov et al., 2018; Feinberg et al., 2019; Brodowsky et al., 2021; Quaglia et al., 2023; Brodowsky et al., 2023) as well
as to evaluate impacts of sulfur-based SAI scenarios (Heckendorn et al., 2009; Vattioni et al., 2019; Weisenstein et al., 2022), which makes them the tools of choice to evaluate SAI of solid particles.

### 2.1 SOCOL-AERv2

SOCOL-AERv2 is based on the chemistry climate model SOCOLv3 (Stenke et al., 2013) which consists of the middle atmosphere version of the spectral general circulation model ECHAM5 (Roeckner et al., 2003, 2006) and the chemistry transport
model MEZON (Rozanov et al., 1999; Egorova et al., 2003). MEZON treats 59 gaseous species of the nitrogen, oxygen, carbon, chlorine, bromine and sulfur families, which are subject to ECHAM5.4 advection. The chemical solver of MEZON is based on the implicit iterative Newton-Raphson scheme (Ozolin, 1992; Stott and Harwood, 1993) and accounts for 16 heterogeneous, 58 photolysis and 160 gas-phase reactions, which represent the most relevant aspects of atmospheric chemistry. The sectional (size resolved) aerosol-microphysics module of the chemistry transport model 2D-AER (Weisenstein et al., 1997, 2007) was then
interactively integrated into the three dimensional grid of SOCOLv3 resulting in the first version of SOCOL-AERv2 (Sheng et al., 2015, i.e., SOCOL-AERv1), which was later further updated by Feinberg et al. (2019, i.e., SOCOL-AERv2). SOCOL-



AERv2 tracks sulfuric acid aerosols within 40 dry aerosol mass bins ranging from 2.8 molecules to $1.6 \times 10^{12}$ molecules corresponding to dry radii from 0.39 nm to 3.2 µm (assuming a density of 1.8 $g/cm^3$) with the number of molecules doubling for subsequent bins. The wet aerosol properties are then calculated in every SOCOL grid box taking into account the $H_2SO_4$
weight percent as a function of relative humidity and temperature (Tabazadeh et al., 1997). AER calculates microphysical processes such as sulfuric acid aerosol formation from gaseous $H_2SO_4$ via nucleation (Vehkamäki et al., 2002) and condensation as well as their evaporation (Ayers et al., 1980; Kulmala and Laaksonen, 1990). Coagulation of sulfuric acid aerosols is calculated using the semi-implicit method of Jacobson and Seinfeld (2004) while the coagulation kernel is calculated using the empirical formula of Fuchs (1964). Finally, sedimentation is treated based on Kasten (1968) adopting the numerical scheme
of Walcek (2000) and aerosols are removed from the model via interactive calculation of wet and dry deposition (Tost et al., 2006, 2007; Kerkweg et al., 2006, 2009; Revell et al., 2018). In the stratosphere, the aerosol module is fully interactive. The aerosol number densities, the wet aerosol volume, the surface area density (SAD) as well as the $H_2SO_4$ weight percent of the aerosols resulting from AER are subsequently passed on to the heterogeneous chemistry scheme and to the RTC of SOCOL-AER, while in the troposphere, prescribed aerosol quantities are used for chemistry and radiative transfer calculations and
aerosol-cloud interactions are not accounted for.

The LW scheme of the RTC of ECHAM5.4 is based on the Rapid Radiative Transfer Model (RRTM, Mlawer et al., 1997) using the correlated k-method with a resolution of 16 bands in the spectral range from 10 $cm^{-1}$ to 3000 $cm^{-1}$. The shortwave code is based on Fouquart and Bonnel (1980) and has a spectral resolution of 6 bands ranging from 185 nm to 4 µm. While the short wave code accounts for scattering, absorption and emission of radiation on aerosols, the RRTM only accounts for
absorption and emission of radiation. Tabulated values of absorption and scattering efficiencies as well as asymmetry factors are used together with the model's aerosol number densities to calculate the scattering and absorption coefficients of the aerosol size distribution, which are then fed into the RTC of SOCOL-AERv2. The tabulated absorption and scattering efficiencies were calculated as a function of aerosol size, $H_2SO_4$ weight percent and spectral resolution based on Mie theory with refractive indexes from Yue et al. (1994) and Biermann et al. (1996).

The version of SOCOL-AERv2 used for this study has a vertical resolution of 39 sigma-pressure levels reaching up to 0.01 hPa (about 80 km altitude) and T42 horizontal resolution ($2.8° \times 2.8°$). The dynamical time step is 15 minutes, while chemistry is calculated every 2 hours. The aerosol microphysics (nucleation, condensation and coagulation) is calculated with operator splitting by applying a loop of 20 iterations within the chemistry call every 2 hours, making the microphysical time step 6 minutes. However, Vattioni et al. (2023c) have shown that for enhanced $H_2SO_4$ supersaturations a microphysical timestep of 6
minutes is not short enough. Therefore, we applied a microphysical timestep of 2 minutes (60 subloops within the chemistry routine) for all $SO_2$ emission scenarios. Other processes relevant for aerosols such as wet and dry deposition and sedimentation as well as calculations of aerosol quantities relevant for radiative transfer and heterogeneous chemistry such as SAD, pH and number densities are calculated and updated every 2 hours.

The same solid particle microphysics module was also incorporated in the fully coupled ESM SOCOLv4 (Sukhodolov
et al., 2021), a further development of SOCOL-AERv2 which is based on the CMIP6 version of MPI-ESM (Mauritsen et al., 2019). While SOCOL-AERv2 and SOCOLv4 share the chemistry and aerosol microphysics scheme, SOCOLv4 is based on



ECHAM6 (Stevens et al., 2013), which incorporates an interactive ocean module (Jungclaus et al., 2013). Furthermore, it provides a finer resolution of the short-wave spectrum as well as a higher spatial resolution and a smaller dynamical timestep, which makes it computationally much more expensive. This paper is based on SOCOL-AERv2, which uses prescribed sea

surface temperatures and sea ice concentrations to study the effective radiative forcing as well as microphysics and impacts on heterogeneous chemistry, while SOCOLv4 will be used in the near future for studies on tropospheric climate impacts of solid particle injections.

## 2.2 The interactive solid particle microphysics module

For the representation of the solid particles, we use a similar sectional approach as for the sulfuric acid aerosols. Particles

are injected as monomers, which can grow to larger order agglomerates via coagulation (see subsections on "Coagulation"). The injected monomer radius can be specified in the model and varies between 80 nm and 320 nm in this study to investigate trade offs between agglomeration, sedimentation speed and backscatter efficiency of different injected monomer radii. To keep track of the monomers and their agglomerates the solid particles are represented by different mass bins (i=1-10), with mass doubling between subsequent bins (i.e., 1-, 2-, 4-, 8-, 16-, 32-, 64-, 128-, 256- and 512-mers). Since coagulation is much more

efficient for smaller particles we only used all 10 solid particle mass bins (up to 512-mer) for injections of particles with small monomer radii, while for radii larger than 200 nm, 5 mass bins (up to 16-mers) are sufficient due to minor agglomeration. The solid particles are fully interactive with the stratospheric sulfur cycle including sulfuric acid aerosols (see subsections on "Coagulation and Condensation"). We also accounted for heterogeneous chemistry taking place on solid particle surfaces (see subsections "Heterogeneous Chemistry") as well as for scattering and absorption of radiation (see subsection "Radiation"),

which makes this the first fully coupled aerosol chemistry climate model to simulate SAI of solid particles except for aerosol cloud interactions. The various processes, which are accounted for in the model are depicted in Figure 1 and described in detail in the following subsections. Since calcite and alumina particles differ significantly in their heterogeneous chemistry and microphysical interactions with sulfuric acid, we present two different model versions for the two particle types. While this Section describes processes which apply to both, calcite and alumina particles (see right part of Figure 1), Sections 2.3 and 2.4

present processes which only apply to alumina and calcite particles, respectively (see left part of Figure 1).

### 2.2.1 Mobility Radius

To represent processes such as sedimentation and coagulation of agglomerates, the mobility radius of the agglomerates ($r_{\mathrm{m,i}}$) is required (Spyrogianni et al., 2018). The mobility radius of alumina ($\rho = 3.98 \, \mathrm{g/cm^3}$) and calcite ($\rho = 2.71 \, \mathrm{g/cm^3}$) agglomerates with primary particle radii of 5 nm, 80 nm, and 215 nm for alumina particles as well 5 nm, 80 nm, and 275 nm for calcite

particles were determined by using a discrete element model (DEM) of particle motion and coagulation (Kelesidis and Kholghy, 2021). The model simulates the coagulation dynamics of nanoparticles and has been validated with experimental data from black carbon (Kelesidis et al., 2017a, b), zirconia (Eirini Goudeli and Pratsinis, 2016) and silica (Kelesidis and Goudeli, 2021) nanoparticles. Furthermore, it was recently interfaced with the discrete dipole approximation (Kelesidis and Pratsinis, 2019; Kelesidis et al., 2020, 2023) and global climate models (Kelesidis et al., 2022) to accurately estimate the direct radiative forcing







**Figure 1.** Schematic illustration of the different processes (yellow boxes) represented in the solid particle microphysics model incorporated in SOCOL-AERv2 and SOCOLv4. The right side of the figure depicts processes relevant for solid particles in general (see Section 2.2) and the left side depicts specific processes relevant to alumina (upper part, see Section 2.3) and calcite (lower part, see Section 2.4) particles. Orange arrows represent most important feedbacks between processes considered in the model.

from black carbon agglomerates. In brief, 1000 monodisperse alumina or calcite particles with initial number concentration of $10^7 - 10^{14}$ cm$^{-3}$ and radii of 5, 80 as well as 215 or 275 nm are randomly distributed in a cubic simulation box at constant pressure and temperature of 50 hPa and 240 K, respectively. Then, the particle motion and coagulation are derived using an event driven method (Goudeli et al., 2015). That way, the evolution of the total number concentration (Section S1, Figure S1)





and size distribution (Figure S2) can be derived accounting for the realistic agglomerate structure. Furthermore, the agglomerate

$r_{m,i}$ can be obtained from its projected area, $A_{proj,i}$ (Rogak et al., 1993):

$$r_{m,i} = \sqrt{\frac{A_{proj,i}}{\pi}} . \tag{1}$$

No significant differences in the resulting average mobility radius of the agglomerates could be observed within the modelled range of initial concentrations (see Figure 2, S1 and S2). The mobility radii of other particle sizes (i.e., 160 nm, 240 nm, and 320 nm, see Section 3) can be linearly extrapolated from the radii resulting for 80 nm, 215 nm, and 275 nm particles.

Further details on the DEM simulations can be found in the supplementary material (Section S1 and Figures S1 and S2). The representation of the particles with the mobility radius is an improvement compared to previous studies (e.g. Weisenstein et al., 2015, who used the radius of gyration assuming the same fractal dimensions of 1.6 or 2.6 for all agglomerates, see Figure S3), especially for representation of sedimentation and thus, the resulting stratospheric aerosol burden. It should be noted that the agglomerate fractal dimension evolves during coagulation and attains its asymptotic value of 1.6-1.8 when agglomerates

containing at least 15 monomers are formed (Goudeli et al., 2015). Thus, assuming constant fractal dimensions can result in an overestimation of the agglomerate number density and mobility radius (see Figure S3 in the SI Kelesidis and Kholghy, 2021).

However, these DEM simulations also showed that it could be challenging to reduce initial particle concentrations in an aircraft wake to levels that are small enough to avoid rapid agglomeration in an aircraft wake (see Figure S2). Most simulations showed agglomerates size distributions peaking at agglomerates between $10^1$ and $10^3$ primary particles per agglomerate after

only two hours, which would reduce scattering efficiencies as well as increase sedimentation speeds of the solid particles. However, these simulations neglected the effect of dilution, which could reduce number concentrations and thus, coagulation. Nevertheless, the neglected injection plume processes at the sub ESM grid scale remain one of the major limitations of most global models including the one used in this study.

### 2.2.2 Sedimentation

The solid particles were integrated into the same sedimentation scheme as applied for sulfuric acid aerosols in SOCOL-AERv2, which is based on Kasten (1968) and Walcek (2000). Following Spyrogianni et al. (2018) we used the mobility radius for calculation of the terminal velocity. The terminal velocity of a falling particle in a fluid can be described with Stokes law, when the Reynolds number is significantly smaller than 1 (Seinfeld and Pandis, 1997). This applies to falling sub-micron particles in the atmosphere. Assuming buoyancy is negligible, the terminal velocity is reached when the drag force ($F_D$, 3) and the

gravitation force ($F_G$, 2) of a falling particle are in equilibrium (i.e., $F_G = F_D$).

$$F_D = \frac{6 \pi \eta_{air} r_{m,i}}{C(r_{m,i})} v_{t,i} \tag{2}$$

$$F_G = m_i \, g = \rho_p \, i \, \frac{4}{3} \, r_0^3 \, g \tag{3}$$

In 3 and 2 $m_i$ is the particle mass of mass bin i, $g$ the gravitational constant, $\eta_{air}$ the viscosity of air, $\rho_p$ the density of the particle, $r_{m,i}$ the mobility radius of the particle, $C(r_i)$ the Cunningham correction of the particles in mass bin i, and $r_0$ the





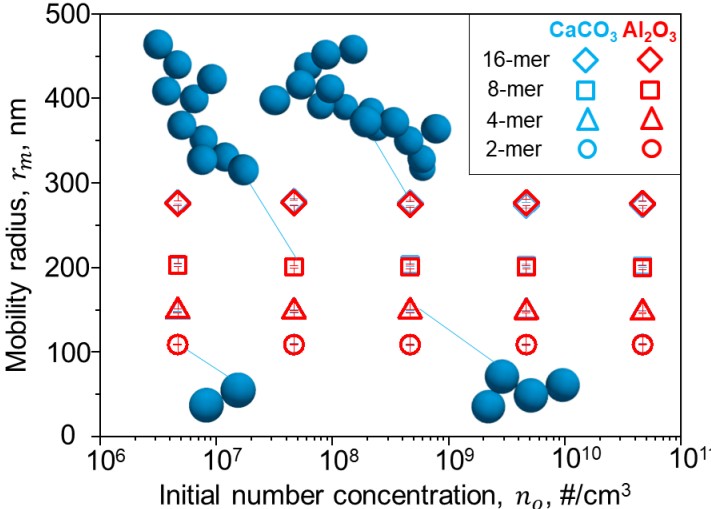

**Figure 2.** The averaged mobility radius of different agglomerates for calcite (blue) and alumina (red) particles with primary particle radius of 5 nm as a function of initial concentrations modelled by a DEM. The averaged shape of agglomerates resulting from initial concentrations of $2 \times 10^{12}$ cm$^{-3}$ is illustrated in dark blue.

monomer radius. Solving for $v_{\text{t,i}}$ gives 4 which is used to calculate the sedimentation speed of the solid particles in the aerosol sedimentation scheme of SOCOL-AERv2 (Feinberg et al., 2019):

$$v_{\text{t,i}} = \frac{m_{\text{i}}\, g\, \text{C}(r_{\text{m,i}})}{6\, \pi\, \eta_{air}\, r_{\text{m,i}}}\ . \tag{4}$$

The resulting sedimentation speeds of solid particles and their agglomerates calculated in SOCOL-AERv2 are shown in the supplementary material in Figure S4.

### 2.2.3 Wet and dry deposition

Solid particles are removed from the atmosphere via the same interactive wet and dry deposition schemes as used for sulfuric acid aerosols in SOCOL-AERv2, which were implemented and tested in Feinberg et al. (2019, see Section 2.1). Uptake of solid particles in cloud and rain droplets is calculated based on a mobility radius-dependent calculation of nucleation and impaction scavenging. Solid particle mass released to the atmosphere after cloud evaporation is added back to the largest available solid particle mass bin. Dry deposition velocities are calculated following the resistance approach by Wesely (1989) using the mobility radius (see Section 2.2.1) and the corresponding densities of the solid particles.



### 2.2.4 Radiation

To make the solid particles interact with the RTC code, SOCOL-AERv2 requires a lookup table with the absorption and scattering efficiencies ($Q_{abs}$ and $Q_{sca}$) normalized to the geometric cross-section obtained from their volume equivalent radii ($r_{ve,i}$) as well as the asymmetry factor ($g_{asy}$) for all mass bins, i. The volume equivalent radius is given by

$$r_{ve,i} = \sqrt[3]{i\,\frac{4}{3}\pi r_0^3}\,, \tag{5}$$

where $r_0$ is the primary particle radius (i.e., monomer radius). While $Q_{abs}$, $Q_{sca}$ and $g_{asy}$ are required for all spectral SW bands, the LW bands only require the look up table for $Q_{abs}$, since the RRTM (Mlawer et al., 1997) incorporated in SOCOL-AERv2 does not account for scattering in the LW spectral bands.

For the monomers these optical properties are calculated from Mie theory utilizing the solution of (Bohren and Huffman, 2008) for calcite and alumina particles, implemented with an open source MATLAB code (Mätzler, 2002). For the aggregates the code developed by Rannou et al. (1999) was applied, which is a semi-empirical fit to a mean-field theory solution of the Maxwell equations for interaction of fractal agglomerates with electromagnetic waves. Both of these codes provide the full scattering phase function, although it is not utilized by SOCOL-AER's RTC. The required inputs for the monomer code are complex refractive index as a function of wavelength and monomer size, which were taken from Tropf and Thomas (1997) for alumina and from Ghosh (1999) and Long et al. (1993) for calcite. This is also consistent with Dykema et al. (2016). For the aggregates, the number of monomers comprising the aggregate and the fractal dimension are also required as an input. Within each SW radiative transfer model band the optical scattering and absorption are weighted by the incident top-of-the-atmosphere (ToA) solar spectrum and averaged, whereas in the LW bands scattering is neglected and absorption is given as a simple average over each spectral band. The resulting $Q_{abs}$ for all spectral bands as well as $Q_{sca}$ and $g_{asy}$ for the SW bands, which were subsequently used in SOCOL-AERv2 are shown in the supplementary material in Figure S5 and Figure S6.

In SOCOL-AERv2 the scattering and absorption cross sections ($\sigma_{sca,i}$ and $\sigma_{abs,i}$) of a particle in mass bin i is given by:

$$\sigma_i = \pi\,r_{ve,i}^2\,Q_i \tag{6}$$

The scattering and absorption coefficients, $\epsilon_{sca,i}$ and $\epsilon_{abs,i}$ of each mass bin are then calculated via equation 7 by multiplying the cross sections of each mass bin with the number densities ($N_i$) of each mass bin. Summing up over all mass bins yields the total scattering and absorption coefficient ($\epsilon_{sca}$ and $\epsilon_{abs}$) for each spectral SW band:

$$\epsilon = \sum_{i=1}^{10}\epsilon_i = \sum_{i=1}^{10}N_i\,\sigma_i \tag{7}$$

The extinction coefficient ($\epsilon_{ext}$) as well as the single scattering albedo ($\omega_{ssa}$) are given by

$$\epsilon_{ext} = \epsilon_{sca} + \epsilon_{abs} \tag{8}$$





$$\omega_{\mathrm{ssa}} = \frac{\epsilon_{\mathrm{sca}}}{\epsilon_{\mathrm{ext}}} \tag{9}$$

Additionally, the bulk asymmetry factor ($g_{\mathrm{asy}}$) for the solid aerosol size distribution is calculated by the sum of each mass bin's assymetry factor ($g_{\mathrm{asy,i}}$) weighted by the corresponding scattering coefficient:

$$g_{\mathrm{asy}} = \frac{1}{\epsilon_{\mathrm{sca}}} \sum_{i=1}^{10} g_{\mathrm{asy,i}} \, \epsilon_{\mathrm{sca,i}} \tag{10}$$

Parameters derived from equations 7 to 10 are then fed to the RTC of SOCOL-AERv2, where the absorption and the scattering due to solid particles is calculated for each spectral band.

For simplicity, the optical properties of the monomers comprising all particles were calculated assuming pure, un-aged materials (i.e., calcite or alumina) for interactions of the particles with radiation, although the model would allow applying optical properties as a function of particle aging if data were available. We used the semi-empirical code of Rannou et al. (1999) to

look at the change of the optical properties of alumina particles with radius of 240 nm when assuming a 10 nm thick spherical sulfuric acid coating (a valid assumption, see Section 4.3), but found only very little changes in scattering and absorption properties (not shown). However, for this calculation an effective medium approximation (i.e., using a volume-weighted function of the refractive indices of the constituent materials, Lesins et al., 2002) was applied to provide an effective refractive index for the alumina-sulfuric acid core-shell. This was necessary because the Rannou et al. (1999) code can only handle homogeneous

spherical constituent monomers. Since composition changes of the particles resulting from the simulations in this study are only small (see Section 4) with only little impact on optical properties, we only accounted for the optical properties of bare calcite and alumina particles. Changes in optical properties as a result of composition changes of the solid particles through aging processes such as uptake of HNO$_3$ on calcite particles resulting in formation of Ca(NO$_3$)$_2$ (see Section 2.4) or as a result of sulfuric acid uptake at the alumina particle surface (see Section 2.3) were not accounted for.

## 2.3 Alumina particles

Alumina particles are represented with two sets of prognostic variables where each set represents 10 mass bins for solid particle monomers and agglomerates as well as an additional prognostic variable for the sulfuric acid coating of each alumina mass bin. One set of mass bins represents particles partially coated by sulfuric acid, while the other set represents particles fully coated by sulfuric acid. This results in a total amount of $4 \times 10$ additional prognostics variables for alumina particle representation, their

agglomerates and their sulfuric acid coating. Particles are emitted as spherical monomers with density $\rho$=3.98 g/cm$^3$ and a molar weight of 101.1 g/mol into the partially coated alumina monomer mass bin. They can acquire a sulfuric acid coating via condensation of H$_2$SO$_{4(g)}$ or via coagulation with sulfuric acid aerosols. When the coating mass per primary particle reaches a certain threshold (see Section 2.3.3, "Contact angle") they are moved to the fully coated mass bins.

### 2.3.1 Coagulation of alumina particles

The coagulation dynamics of solid particles and their interaction with sulfuric acid aerosols were integrated in the same semi-implicit coagulation schemes for sulfuric acid aerosols presented in Sheng et al. (2015) and Feinberg et al. (2019, see subsec-





tion "SOCOL-AERv2"). Coagulation of solid particles and sulfuric acid aerosols are represented following the description in Weisenstein et al. (2015, see their Appendix A), with the only difference that here we do not account for pure solid particles and coated solid particles, but for partially coated and fully coated particles (see Figure 1, alumina particles). We account
for self coagulation of sulfuric acid aerosols, partially coated solid particles and fully coated sulfuric acid particles as well as coagulation between these three categories.

As already stated in Weisenstein et al. (2015) applying a discrete aerosol mass binning leads to an artificial broadening of the particle size distribution since coagulation would often result in agglomerates of sizes, which fall in between two mass bins. In these cases the resulting mass is split up between neighboring mass bins by applying a statistical weighting Weisenstein et al.
(see 2015, Appendix A). The resulting error depends on the coarseness of the bin spacing (Weisenstein et al., 1997, 2007), which is a mass doubling in the presented model. This is a good compromise between accuracy of representation and usage of computational resources (Weisenstein et al., 2015), since computational resources increase with every additional prognostic variable.

The coagulation kernel was calculated using the mobility radius for solid particle agglomerates (see previous section "Mo-
bility radius") and the spherical radius for liquid sulfuric acid aerosols for every possible combination of collisions i.e., self coagulation between aerosol mass bins of every category (40 liquid sulfuric acid mass bins, 10 partially coated and 10 fully coated solid particle mass bins) as well as coagulation between all mass bins of every aerosol particle category. The calculation of the coagulation kernel followed the same methodology as for sulfuric acid aerosols in SOCOL-AERv2 (see Section 2.1) and was implemented following (Weisenstein et al., 2015, Appendix A). The representation applied only accounts for Brownian
coagulation and neglects gravitational, convective and Van der Waals corrections, which results in slight underestimation of coagulation efficiencies.

### 2.3.2 Condensation and evaporation of $H_2SO_4$

Condensation of $H_2SO_4$ on alumina particles and evaporation of $H_2SO_4$ from alumina particles was treated the same way as described in Weisenstein et al. (2015, Appendix A) following the methodology described in Jacobson and Seinfeld (2004).
However, contrary to Weisenstein et al. (2015) we accounted for condensation of $H_2SO_4$ not only on fully coated, but also on partially coated solid particles. The $H_2SO_4$ condensation rates on solid particles are calculated as a function of the SAD of solid particles, their number density, the molecular diffusion coefficient of $H_2SO_4$, the difference between the $H_2SO_4$ partial pressure and the $H_2SO_4$, equilibrium vapour pressure as well as the primary particle radius of every agglomerate to account for the Kelvin effect. Evaporation of $H_2SO_4$ is represented with the same scheme as for condensation and occurs when the partial
pressure of $H_2SO_4$ is smaller than the equilibrium vapour pressure of $H_2SO_4$, which mainly occurs above 35 km altitude.

### 2.3.3 Contact angle of $H_2SO_{4(aq)}$ on solid particles

For partially coated alumina particles, the sulfuric acid coating is represented by accounting for the contact angle ($\theta$) of $H_2SO_4 - H_2O$ on alumina particles to differentiate between surface area covered by sulfuric acid and uncovered $Al_2O_3$ surface area. Figure 3 a) shows the basic geometry of a partial sphere from which equations were derived to calculate the share of the



two types of surface area (Polyanin and Manzhirov, 2006). The volume of liquid sulfuric acid per monomer ($V_{\text{liq}}$) as well as the contact angle ($\theta$) is known and $\beta$ can then be determined by inserting equations 12-16 into equation 11 (see Figure 2c). In equation 11 $V_{\text{p}}$ and $V_{\text{l}}$ are the volumes of the partial spheres of the solid particle and the liquid sulfuric acid respectively (see Figure 3a), while $h$ is referring to the height of the missing part of the sphere, $r$ to the radius of the partial spheres and $c$ to the base radius of the partial spheres (see Figure 3a) of liquid sulfuric acid (l) and the solid particle (p, see Figure 3c).

$$V_{\text{liq}} = V_{\text{l}} - V_{\text{p}} = \frac{\pi}{6}\,h_{\text{l}}\,\left(3c^2 + h_{\text{l}}^2\right) - \frac{\pi}{6}\,h_{\text{p}}\,\left(3c^2 + h_{\text{p}}^2\right) \tag{11}$$

$$h_{\text{l}} = r_{\text{l}} - r_{\text{l}}\,\left(\cos\left(\theta + \beta\right)\right) \tag{12}$$

$$h_{\text{p}} = r_{\text{p}} - r_{\text{p}}\,\left(\cos\left(\beta\right)\right) \tag{13}$$

$$c = r_{\text{p}}\,\sin\left(\beta\right) \tag{14}$$

$$c = r_{\text{l}}\,\sin\left(\theta + \beta\right) \tag{15}$$

$$r_{\text{l}} = \frac{\sin\left(\beta\right)\,r_{\text{p}}}{\sin\left(\theta + \beta\right)} \tag{16}$$

The solid particle surface area and the sulfuric acid surface area per solid particle monomer can then be calculated with 17 and 18.

$$S_{\text{liq}} = \pi\,\left(c^2 + h_{\text{l}}^2\right) \tag{17}$$

$$S_{\text{solid}} = 4\,\pi\,r_{\text{p}}^2 - \pi\,\left(c^2 + h_{\text{p}}^2\right) \tag{18}$$

The liquid sulfuric acid volume of each mass bin is assumed to be equally distributed over all primary particles within one agglomerate assuming that every primary particle hosts the same amount of sulfuric acid. The whole alumina and sulfuric acid coating mass is transferred to the fully coated mass bins as soon as $\beta$ is larger than 90°, an arbitrarily but realistic criteria for immersion (see Figure 3b). The fully coated mass bins assume the alumina particles to be equally spherical and fully covered by sulfuric acid (see Figure 1, "Alumina particles").

### 2.3.4 Heterogeneous chemistry on alumina particles

The sulfuric acid SAD resulting from the partially coated alumina particles as well as the one from fully coated alumina particles is added to the total available sulfuric acid aerosol SAD and the same heterogeneous chemistry is assumed to take place on this surface area as for sulfuric acid aerosols (Sheng et al., 2015). On alumina SAD of partially coated alumina particles, we accounted for R1, R2 and R3:

$$\text{ClONO}_2 + \text{HCl} \;\rightarrow\; \text{Cl}_2 + \text{HNO}_3 \tag{R1}$$

$$\text{ClONO}_2 + \text{H}_2\text{O} \;\rightarrow\; \text{HClO} + \text{HNO}_3 \tag{R2}$$

$$\text{N}_2\text{O}_5 + \text{H}_2\text{O} \;\rightarrow\; 2\text{HNO}_3 \tag{R3}$$





**Figure 3.** Schematic illustration of the representation of the contact angle of $H_2SO_4$ (depicted in orange) on alumina particles (depicted in grey). Panel a) depicts the general geometry of a partial sphere with basic equations. The criteria for immersion is illustrated in panel b). An additional molecule of $H_2SO_4$ acquired on this particle will lead to transfer of the particle mass to the fully coated mass bins. Panel c) illustrates the quantities used for the equations used in the main text to determine the angle $\beta$ (see 11-16), which is then used to determine the sulfuric acid SAD and the alumina SAD (see 17 and 18).

Though Molina et al. (1997) measured uptake coefficients for reaction R1 (R1), their data are not representative for low stratospheric HCl partial pressures. To extrapolate the experimental data of Molina et al. (1997) to typical stratospheric values, we applied a Langmuir-Hinshelwood representation of adsorption and reaction as detailed in Vattioni et al. (2023b). For the this study we used the scenario "dissociative $\gamma$, $\alpha$=0.1" to calculate the uptake coefficient of ClONO$_2$ on alumina particles for





R1. Due to the lack of experimental data on other heterogeneous reactions we only accounted for R2 and R3 by assuming the same reaction rates as on sulfuric acid aerosols, which is an upper limit estimate (Vattioni et al., 2023b).

### 2.4 Calcite particles

In contrast to alumina, calcite is alkaline and thus reactive towards the acids in the stratosphere. Therefore, calcite particle can change their composition by forming salts at the surface (Keith et al., 2016; Cziczo et al., 2019; Huynh and McNeill, 2020; Dai et al., 2020; Huynh and McNeill, 2021). This requires a different treatment than for alumina particles, which do not undergo compositional changes, but only acquire a sulfuric acid coating at the surface.

#### 2.4.1 Heterogeneous chemistry on calcite particles

For calcite particles the following heterogeneous reactions upon uptake of HCl, HNO$_3$ and H$_2$SO$_4$ are considered (R1-R3):

$$CaCO_3 + 2HCl \rightarrow CaCl_2 + H_2O + CO_2 \tag{R4}$$

$$CaCO_3 + 2HNO_3 \rightarrow Ca(NO_3)_2 + H_2O + CO_2 \tag{R5}$$

$$CaCO_3 + H_2SO_4 \rightarrow CaSO_4 + H_2O + CO_2 \tag{R6}$$

To keep track of the reaction products (Ca(NO$_3$)$_2$, CaCl$_2$ and CaSO$_4$) additional prognostic variables for all three products

were implemented for every calcite mass bin, resulting in a total of 40 prognostic variables (4 species times 10 mass bins). The total number of molecules per particle is always the same, but depending on the uptake of acids they are either in the form of CaCO$_3$, Ca(NO$_3$)$_2$, CaCl$_2$ and CaSO$_4$. This changes the density of the particles (i.e., $\rho_{CaCO_3}$=2.71 g/cm$^3$, $\rho_{CaSO_4}$=2.32 g/cm$^3$, $\rho_{Ca(NO_3)_2}$=2.50 g/cm$^3$, $\rho_{CaCl_2}$=2.15 g/cm$^3$) and therefore also their radius, which is accounted for in the model. As stated by Cziczo et al. (2019) this is a simplification since in reality the reaction products would form hydrates, which are less

dense then their anhydrous forms, and likely also mixed salts.

R4-R6 are treated as first order reactions resulting in the following mass balance for calcite and the reaction products:

$$\frac{d[CaCO_3]}{dt} = -0.5[HCl]k_{HCl+CaCO_3} - [H_2SO_4]k_{H_2SO_4+CaCO_3} - 0.5[HNO_3]k_{HNO_3+CaCO_3} \tag{19}$$

$$\frac{d[CaCl_2]}{dt} = 0.5[HCl]k_{HCl+CaCO_3} \tag{20}$$

$$\frac{d[Ca(NO_3)_2]}{dt} = 0.5[HNO_3]k_{HNO_3+CaCO_3} \tag{21}$$

$$\frac{d[CaSO_4]}{dt} = [H_2SO_4]k_{H_2SO_4+CaCO_3} \tag{22}$$

Values in brackets are the molecule number densities of the different species. The resulting CO$_2$ and H$_2$O from reactions R4

to R6 is not further tracked since resulting quantities are very small compared to background concentrations of these species. For the calculation of heterogeneous chemistry CaCO$_3$ molecules of all bins are summed up, but the resulting products are



redistributed to the different size bins depending on the share of available SAD from each mass bin. The SAD is always assumed to be pure CaCO$_3$, which means that all reaction sites are always available for reaction. Therefore, no passivation occurs, but instead a constant uptake coefficient ($\gamma$) is applied to calculate the reaction rate ($k$) for reaction R4-R6 following equation 23, where $\overline{v}$ is the thermal velocity of the molecule colliding with the surface (i.e., HCl, HNO$_3$ or H$_2$SO$_4$ in this case):

$$k = \frac{\gamma \, \overline{v} \, \mathrm{SAD}}{4} \tag{23}$$

For simplicity, we neglect temporal variation in the uptake coefficients. Therefore, the passivation effect of the surface must be accounted for via the uptake coefficient $\gamma$, which should be representative for the whole stratospheric lifetime of the calcite particles (about 1 year) and not only for the generally much larger initial reactive uptake on pure calcite particles such as measured in Huynh and McNeill (2020, 2021). The setup presented here allows for sensitivity analysis of different processes such as varying the uptake coefficients and analyzing the total uptake of HCl, HNO$_3$ and H$_2$SO$_4$ as well as the impact on stratospheric chemistry.

In this study we applied uptake coefficients of $10^{-4}$ and $10^{-5}$ for the uptake of HNO$_3$ (R5) and HCl (R4), respectively, following Dai et al. (2020), and an uptake coefficient of 1.0 for H$_2$SO$_4$ (R6), assuming that every collision of a H$_2$SO$_4$ molecule with a calcite particle results in immediate uptake and reaction to CaSO$_4$. Other heterogeneous chemistry on calcite particles is neglected.

### 2.4.2 Coagulation of calcite particles

Coagulation of calcite particles is calculated by the same schemes as for alumina particles. However, instead of tracking the sulfuric acid coating, the CaCO$_3$, Ca(NO$_3$)$_2$, CaCl$_2$ and CasO$_4$ mass per bin is tracked. Additionally, coagulation of calcite particles with sulfuric acid aerosols is assumed to result in instantaneous and irreversible formation of CaSO$_4$ (same as reaction R6).

## 3 Experimental setup

Each injection scenario (see Table 1) emitted continuously between 30°S and 30°N at all longitudes at 54 hPa (∼20 km altitude). The baseline scenarios injected alumina and calcite particles at particle radii of 240 nm at a rate of 5 Mt/yr. Additionally, we performed sensitivity analyses with respect to the injected particle radius, the injection rate, as well as the sulfuric acid contact angle on alumina particles (see Table 1 for details). For comparison with sulfur-based SAI, different scenarios with injections of gaseous SO$_2$ as well as accumulation-mode aerosol of condensed H$_2$SO$_4$, assuming a log-normal distribution with a mean radius of 0.095 µm and a $\sigma$ of 1.5, were also simulated (see Vattioni et al., 2019; Weisenstein et al., 2022, see Table 1 for details). The latter scenario assumes that an aerosol size distribution with a mean radius of 0.095 µm can be produced by injecting gaseous H$_2$SO$_4$ into an aircraft plume (Pierce et al., 2010; Benduhn et al., 2016; Vattioni et al., 2019; Weisenstein et al., 2022). The resulting aerosol size distribution could result in larger radiative forcing (RF), while simultaneously reducing





some side effects such as ozone depletion compared to $SO_2$ injections. However, the underlying assumptions are subject to large uncertainty (Vattioni et al., 2019).

All simulations are time-slices spanning 20 years with all boundary conditions set to the year 2020. For sea surface temperatures (SST) and sea ice concentrations (SIC), a climatological 10-year (2010-2019) average seasonal cycle from the Hadley dataset was used (Kennedy et al., 2019), while concentrations of GHG and ozone depleting substances (ODS) were taken from SSP5-8.5 (O'Neill et al., 2015) and WMO (2018), respectively. The first 5 years of each simulation served as spin-up to equilibrate stratospheric aerosol burden. Hence, all SOCOL-AERv2 data shown in this study are 15-year averages. The boundary conditions follow the GeoMIP test-bed experiment "accumH2SO4"[1] except for injecting the absolute mass of each species and not the equivalent sulfur mass as well as for the boundary conditions following the year 2020 instead of 2040 (see also Weisenstein et al., 2022).

**Table 1.** Overview of the simulations performed in this study. Columns show the emitted species, the injection rate, the injected primary particle radius as well as the contact angle where applicable. Injections were emitted continuously between 30°N and 30°S at 20 km altitude. The baseline configurations are marked in bold.

| Emitted Species | Injection Rate ($Mt\ yr^{-1}$) | Injected Primary Particle Radius | Contact Angle |
|---|---|---|---|
| Alumina | 1, **5**, 10 and 25 | **240 nm** | 15°, **30°**, 45°, 60°, fully covered |
| Alumina | **5** | 80 nm, 160 nm, **240 nm**, 320 nm | **30°** |
| Calcite | 1, **5**, 10 and 25 | **240 nm** | n/a |
| Calcite | **5** | 80 nm, 160 nm, **240 nm**, 320 nm | n/a |
| $SO_2$ | 1, **5**, 10 and 25 | n/a | n/a |
| AM-$H_2SO_4$ | 1, **5**, 10 and 25 | $r$=0.95 μm, $\sigma$=1.5 | n/a |

## 4 Results

The stratospheric sulfur cycle is usually represented with sulfur equivalent burden (i.e., Gg S), fluxes and injection rates (i.e., GgS/yr) in both, SAI and non-SAI studies (e.g. Feinberg et al., 2019; Weisenstein et al., 2022; Brodowsky et al., 2023). This allows easy comparison of burden and fluxes of different sulfur species. However, when comparing SAI scenarios with gaseous (e.g., $SO_2$), liquid (e.g., $H_2SO_4 - H_2O$, i.e., sulfuric acid aerosols) and solid (e.g., $CaCO_3$ and $Al_2O_3$) species to each other, it is important to compare both the absolute burden and injection rates to allow for direct comparison (see Figure 4). Thus, compared to the sulfur equivalent burden the resulting $H_2SO_4 - H_2O$ burden is larger by a factor of about 3 when accounting for $H_2SO_4$ plus another 40-50% when accounting for the aerosol water content. Therefore, the resulting sulfuric acid aerosol burden reported in Figure 4a are much larger compared to previous studies (e.g., Weisenstein et al., 2015), which compared the solid particle burden and injection rates to sulfur equivalent quantities without accounting for $H_2O$. The comparison shown

---

[1]Details of the experiment protocol: http://climate.envsci.rutgers.edu/geomip/testbed.html



in Figure 4a shows that, for a given injection rate, the resulting sulfuric acid burden is about a factor of $\sim 2$ larger compared to the burden resulting from calcite and alumina particle injections. This is mainly due to the larger densities (i.e., 1.69 $\mathrm{g/cm^3}$ for 70 wt% $H_2SO_4$, 2.71 $\mathrm{g/cm^3}$ for $CaCO_3$ and 3.95 $\mathrm{g/cm^3}$ for $Al_2O_3$) as well as the larger particle radius for the solids, which makes them sediment much faster.

The resulting globally averaged alumina particle burden for an injection of 5 Mt/yr of 80 nm, 160 nm, 240 nm and 320 nm particles are 5.6 Mt, 4.7 Mt, 3.8 Mt and 3.0 Mt, respectively, and therefore about one third smaller compared to the ones found in Weisenstein et al. (2015). This is likely not a result of differences in sedimentation speeds between the models since our modelled sedimentation velocities are slightly smaller compared to the ones shown in Weisenstein et al. (2015, see Figure S3) despite applying different representations of the agglomerate particle radius (see Section 2.2.1). However, compared

to the original 2D-AER code used in Weisenstein et al. (2015) SOCOL-AERv2 has undergone several updates (e.g., Sheng et al., 2015; Feinberg et al., 2019; Vattioni et al., 2023c). Most notably, updates include the replacement of the simple updraft sedimentation scheme by the numerical scheme of Walcek (2000) to reduce numerical diffusion, implementation of interactive wet and dry deposition schemes and updates to the coagulation kernel. The difference in burden might also be affected by the three dimensional representation of dynamics and transport in our model compared to the 2D-zonal mean representation in

2D-AER. The large number of differences between the two models make it difficult to identify which specific processes are responsible for the differences in results.

### 4.1 Radiative Forcing Efficiency

For the same injection rates we find that $AM-H_2SO_4$ injections result in the largest net all sky ToA RF, slightly larger than $CaCO_3$ injections of 240 nm radius. Injecting $SO_2$ results in similar net ToA all sky RF as for $AM-H_2SO_4$ for injection

rates of 10 Mt/yr and smaller. At very large injection rates of 25 Mt/yr, a non-linearity in the RF efficiency of $SO_2$ injections becomes apparent; $SO_2$ injections result in smallest net ToA all sky RF values compared to injections of the other species investigated in this study. This is mainly due to the unfavourable aerosol size distribution resulting from the large continuous $H_2SO_4$ condensation fluxes at large $SO_2$ injection rates, which shifts the aerosol size distribution towards larger particles, which decreases the total scattering cross section per resulting aerosol burden (Heckendorn et al., 2009; Vattioni et al., 2019;

Weisenstein et al., 2022). The injection of $Al_2O_3$ particles of 240 nm radius results in about 25% less net ToA all sky RF compared to injections of $AM-H_2SO_4$ and $CaCO_3$ particles with radii of 240 nm across all the investigated injection rates. However, both the injection of $Al_2O_3$ and $CaCO_3$ particles result in larger RF per unit of stratospheric aerosol burden compared with the sulfur based injection scenarios. Despite the larger aerosol burden in our model, the resulting net ToA all sky RF shown in Figure 4b is in agreement with the net clear sky RF values found in Weisenstein et al. (2015). However, the largest net all sky

ToA RF is achieved with SAI of particles with 160 nm radius, which is in contrast with Weisenstein et al. (2015) for alumina particles, where the largest RF was obtained for injection of slightly larger particles of 240 nm radius.



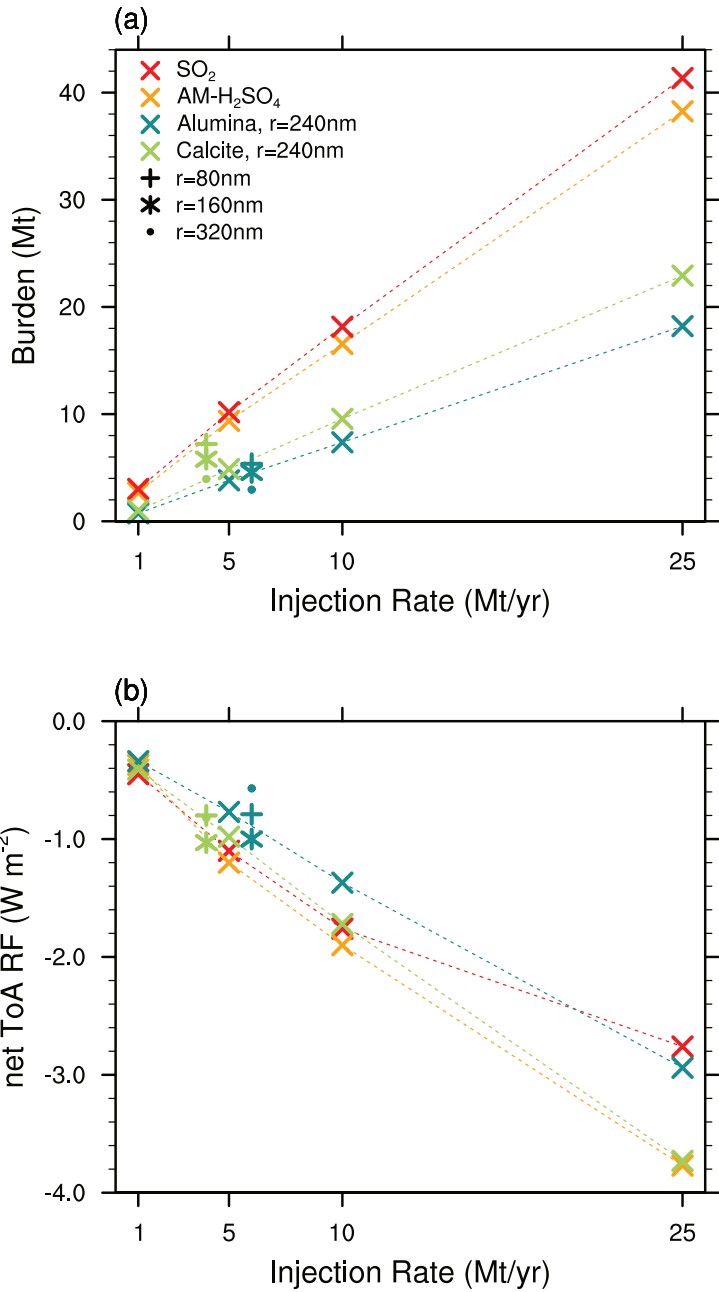

**Figure 4.** Resulting stratospheric aerosol burden (a) and total net all sky ToA RF (b) as a function of injection rate. Shown are absolute injection rates (i.e. Mt $H_2SO_4$/yr and $SO_2$/yr and not Mt S/yr) and absolute burden (i.e., the wet sulfuric acid burden in Mt $H_2SO_4 - H_2O$ and not Mt S).





## 4.2  Coagulation

Both the resulting stratospheric burden as well as the RF scale almost linearly with increasing injection rate for solid particles (see Figure 4). This linearity is mainly due to the relatively small agglomeration found for the injected 240 nm particles even at large injection rates (see Figure 5). This is likely different in scenarios which inject smaller particles (e.g., $r$=80 nm) or which apply larger injection rates and more confined injection regions. The fraction of monomers for injections of 5 Mt/yr of 80 nm, 160 nm, 240 nm and 320 nm particles amounts to 13%, 48%, 82% and 92%, respectively, which is slightly more than what was found in Weisenstein et al. (2015). The more efficient formation of agglomerates in Weisenstein et al. (2015) could be due to the different representation of the radius of the agglomerates (see Section 2.2.1) or updates in the coagulation scheme (see subsection "Coagulation"). Aerosol size distributions for the scenarios injecting 5 Mt/yr of particles with radius of 240 nm can be found in the supplement in Figure S7. For the model presented here, the only scenario resulting in significant agglomeration is the one injecting particles at 80 nm radius, where most of the particle mass is in the form of 16-mers (i.e., mass bin 5, see Figure 5). However, these results are subject to large uncertainties due to lack of resolution of sub-ESM grid scale plume injection processes (Blackstock et al., 2009). In the injection plume (e.g., of an aircraft) the particle concentrations would be significantly higher, which could result in effective agglomeration, whereas we only assume injections equally distributed to the grid of the climate model (i.e., about 325 km × 325 km × ~1.5 km in SOCOL-AERv2 at the equator at 50 hPa).

## 4.3  The Stratospheric Sulfur Cycle under conditions of SAI of alumina particles

Previous studies showed that injection of solid particles will likely result in uptake of sulfuric acid at the particle surface via coagulation with sulfuric acid aerosols and via condensation of gaseous sulfuric acid (Weisenstein et al., 2015; Keith et al., 2016). These processes are also represented in the model presented here (see Figure 6). On the one hand, injecting 5 Mt per year of alumina particles will deplete the global stratospheric background sulfuric acid layer mass by 86%, 69%, 54% and 45% for injection of 80 nm, 160 nm, 240 nm and 320 nm particles, respectively (see Figure 6). On the other hand, the mass of sulfuric acid coating on alumina particles reaches values of 78%, 53%, 35% and 24% of the unperturbed global stratospheric sulfuric acid aerosol burden, respectively (see Figure S7 in the supplement for resulting aerosol size distributions). The sum of the globally averaged stratospheric coating and sulfuric acid aerosol mass are smaller than the unperturbed stratospheric sulfuric acid aerosol burden, which is due to the faster removal via sedimentation of condensed sulfuric acid mass on heavier solid particles compared to sulfuric acid aerosols. Injection of 80 nm particles results in the largest coating mass of sulfuric acid; this is mainly due to the larger coagulation efficiency with sulfuric acid aerosols of small particles, as well as due to the larger surface area availability for condensation. The bigger fraction of sulfuric acid coating is acquired via direct condensation of $H_2SO_{4(g)}$ in all scenarios. However, the share of acquisition via coagulation increases with decreasing alumina particle size from 18% for 320 nm particle injection to 42% for 80 nm particle injection. The same tendencies in the response of the stratospheric sulfur cycle to alumina injection can be observed when increasing the injection rate from 1 Mt/yr to 25 Mt/yr (see supplementary material, Figure S8).



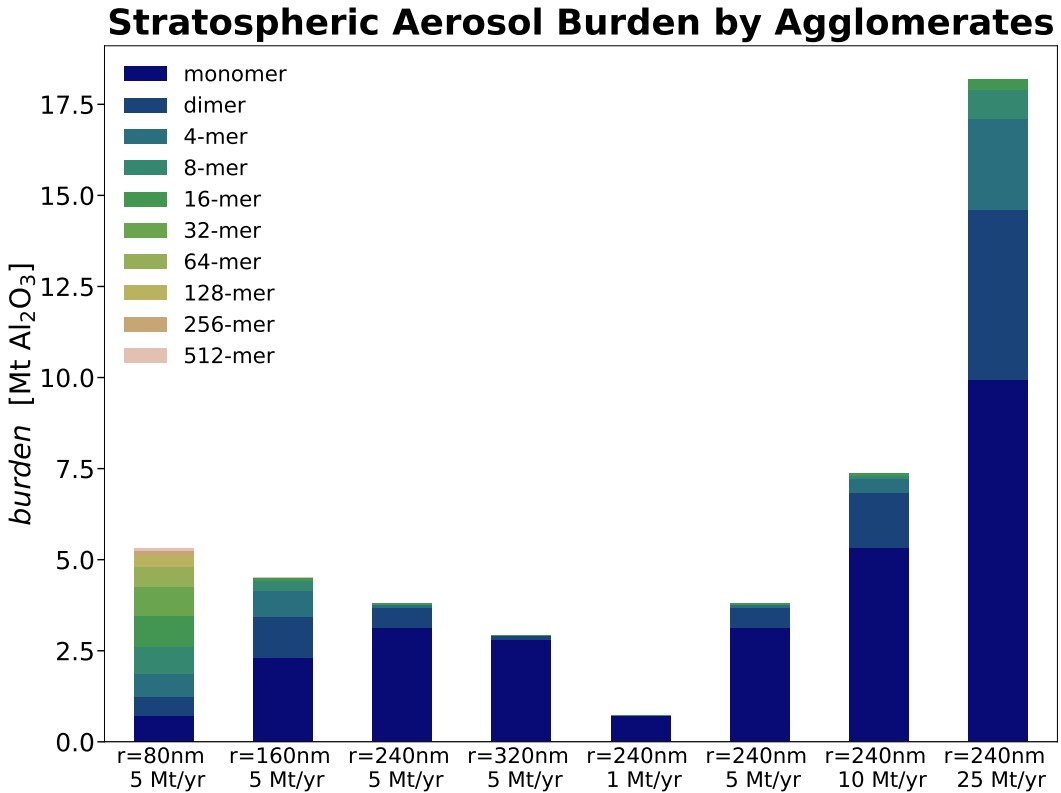

**Figure 5.** The resulting globally averaged stratospheric aerosol burden resolved for the contribution of the individual mass bins resulting from 5 Mt/yr injection of 80 nm, 160 nm 240 nm and 320 nm particles as well as 1 Mt/yr, 5 Mt/yr, 10 Mt/yr and 25 Mt/yr injection of 240 nm particles.

When distributing the sulfuric acid coating (i.e., the total condensed $H_2SO_4 - H_2O$ mass on the alumina particles) equally
on the alumina particles the corresponding coating thickness would reach values of maximal 6-10 nm, 4-8 nm, 7-14 nm and 8-16 nm for injections of 5 Mt/yr of 80 nm, 160 nm, 240 nm and 320 nm particles in the lower stratosphere (see Figure 7). Similar coating thicknesses can be found for different injection rates of particles with radius of 240 nm (see figure S9 in the supplementary material).

## 4.4 Contact Angle Sensitivity Analysis

The sulfuric acid coating thickness on alumina particles shown in Figure 7 is only representative if the sulfuric acid coating is distributed uniformly on the alumina particle surface, which is likely not true for the real system. In Vattioni et al. (2023b) we have performed contact angle measurements of $H_2SO_4$ at different weight percentages and we found a contact angle of about $31° \pm 7°$ at 70 wt% $H_2SO_4$. This measurement is subject to large uncertainty, since the contact angle is dependent on factors such as the relative humidity and the temperature during the measurement as well as the surface characteristics



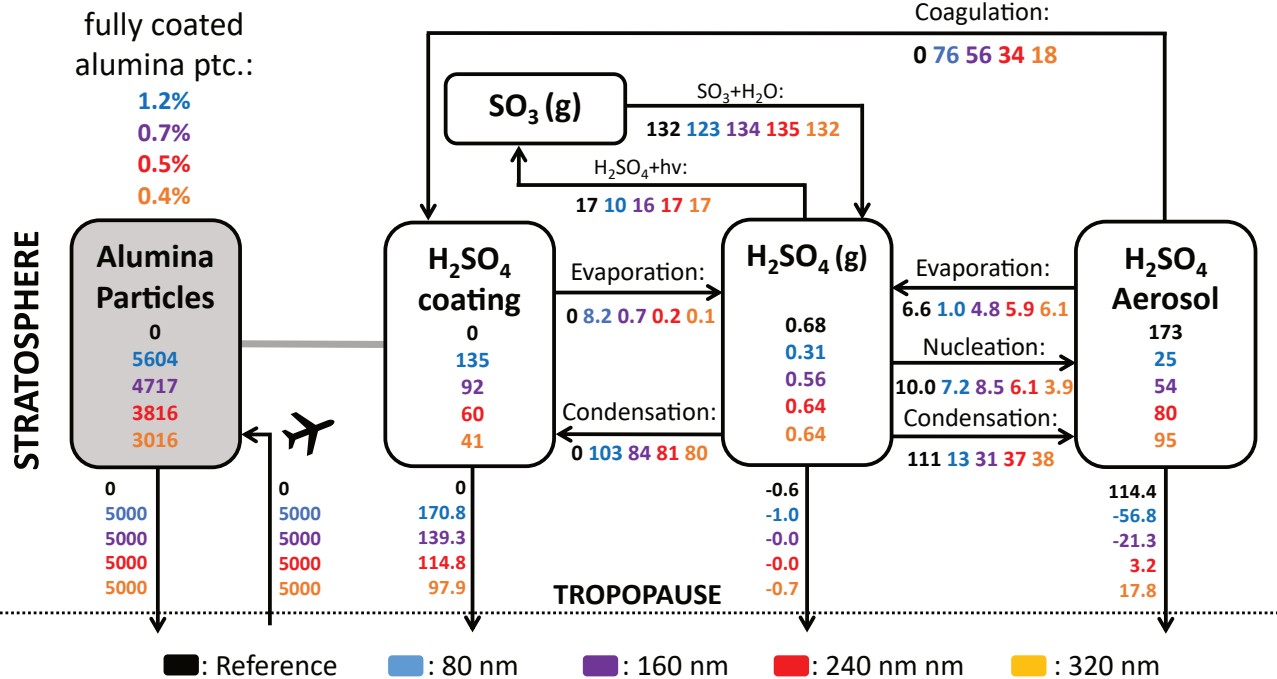

**Figure 6.** The stratospheric sulfur cycle under conditions of SAI of 5 Mt/yr alumina particles with radii of 80 nm (blue), 160 nm (violet), 240 nm (red) and 230 nm (orange). All sulfur species are shown in Gg Sulfur for burden (boxes) and Gg Sulfur per year for net fluxes (arrows). The alumina burden (gray box) is given as Gg $Al_2O_3$ and fluxes as Gg $Al_2O_3$ per year. Cross tropopause fluxes are calculated by balancing the mass balance of the individual species. The percentages in the upper left of the figure indicates the share of fully coated alumina particles for each scenario.

(polished vs. unpolished, cleaned vs. uncleaned). However, the results show that $H_2SO_4$ is likely contracting on alumina surfaces, which would leave parts of the alumina surface uncovered from $H_2SO_4 - H_2O$. Therefore, the sulfuric acid coating on alumina particles is represented by accounting for the contact angle in the model presented here (see Section 2.3.3). We have performed sensitivity simulations on the stratospheric ozone response from applying contact angles ranging from 15° to 60° as well as assuming the alumina particles to be fully coated by sulfuric acid (see Figure 8).

This sensitivity analysis shows that particles assumed to be fully covered with sulfuric acid lead to smallest impacts on stratospheric ozone. This is mostly due to the relatively small resulting total SAD of alumina particles when injecting 5 Mt/yr of particles with 240 nm radius (Figure 10e). Depletion of background sulfuric acid aerosol SAD, which consist mostly of much smaller particles (size distribution peaking at $r$=80 nm, see Figure S7 in the supplement) is compensated by the additional alumina SAD covered by sulfuric acid. In the case of 1 Mt/yr injections, this reduces the overall sulfuric acid SAD and thus even

results in an increased global mean TOC. However, as discussed previously complete coverage of alumina particle by sulfuric acid is unlikely. Therefore, it is more realistic to assume representation of sulfuric acid coating with the contact angle of

**Figure 7.** The resulting coating thickness when injecting 5Mt/yr of alumina particles with radii of 80 nm (a), 160 nm (b), 240 nm (c) and 320 nm (d). The values listed above correspond to the average coating thickness of the mass bin with the largest share of alumina burden (i.e., bin 5 for 80 nm particle injection and bin 1 for the others).



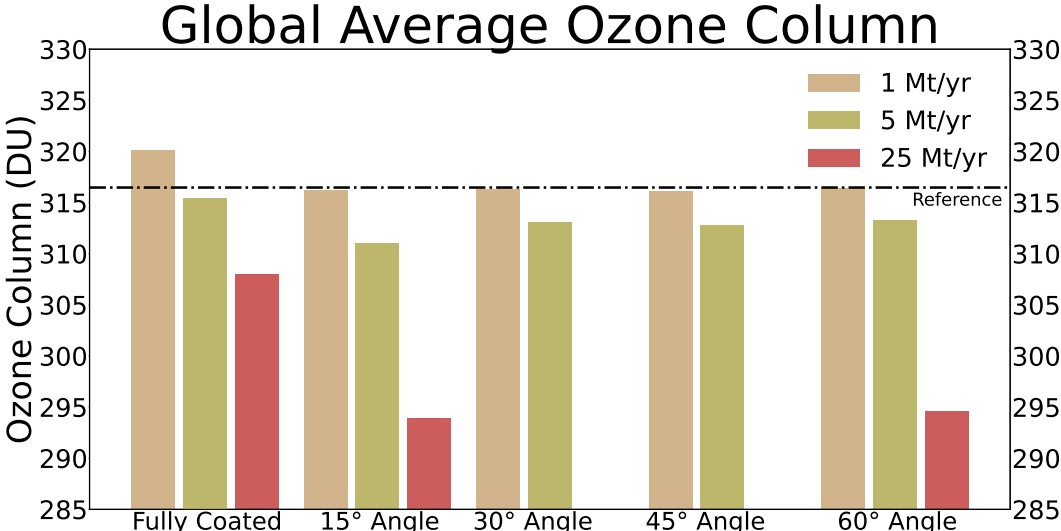

**Figure 8.** The resulting globally averaged total ozone column from 1 Mt/yr, 5 Mt/yr, 10 Mt/yr and 25 M/yr alumina injections when applying a $H_2SO_4-H_2O$ contact angle of 15°, 30°, 45° and 60° as well as when assuming the alumina particles to be fully covered by sulfuric acid (colored bars). Black line shows the total ozone column of the reference scenario.

$H_2SO_4-H_2O$ on alumina surfaces. Applying a contact angle between 15° and 60° leads to higher ozone depletion, mainly due to the availability of uncoated alumina surface and the resulting chlorine activation (see Subsection "Heterogeneous chemistry on alumina particles"). However, there are no significant differences between different contact angles (Figure 8) since for all
cases at least 50% of the alumina surface will remain uncovered by sulfuric acid and the sulfuric acid coating SAD does not significantly change when represented with contact angles of 15° or 60°. Therefore, we use the measured contact angle of 30° for the representation of the sulfuric acid coating on alumina particles. This is a valid assumption given the small coating thickness.

### 4.5 The Stratospheric Sulfur Cycle under conditions of SAI of calcite particles

For the injection of calcite particles, the depletion of the background sulfuric acid aerosol layer as well as condensation and coagulation fluxes on calcite particles are very similar compared to the injection of alumina particles (Figure 9). The only difference compared to alumina particles is that sulfuric acid on calcite particles is immediately assumed to undergo irreversible reaction with $CaCO_3$ to form $CaSO_4$. The resulting globally averaged $CaSO_4$ burden varies between 90 Gg and 296 Gg for 80 nm and 320 nm particles, respectively. This corresponds to only 4.1 % and 2.3% of the entire stratospheric calcite burden,
respectively. At the same time the uptake of HCl with a uptake coefficient of $10^{-5}$ results in $CaCl_2$ burden of 2 Gg and 24 Gg for 320 nm and 80 nm particle injections respectively, which is 0.05% and 0.3% of the resulting total globally averaged stratospheric calcite burden. The biggest fraction other than $CaCO_3$ comes from calcium nitrate, which results from uptake of $HNO_3$ at an uptake coefficient of $10^{-4}$. $Ca(NO_3)_2$ burden are between 65 Gg and 456 Gg for 80 nm and 320 nm particle





injection, respectively, accounting for 1.6% and 6.3% of the resulting total globally averaged stratospheric calcite burden, respectively (see Section S6 and Figure S10 in the supplement for sensitivity to injection rate). Therefore, between 89% and 96% of the calcite burden will remain unchanged in the form of $CaCO_3$ during the entire stratospheric residence time for injection of 80 nm particles and 320 nm particles, respectively. Thus, the scattering and absorption properties of the calcite particles are unlikely to change significantly due to ageing processes. However, the ageing has significant consequences for heterogeneous chemistry on the particle surfaces, since these salts might host different heterogeneous reactions at different reaction rates. The sensitivity analysis of the role of heterogeneous chemistry of calcite particles using this model will be topic of another publication.

## 4.6  Solid particle number concentrations and surface area densities

The resulting solid particle number concentrations reach values of up to 7 particles per $cm^3$ in the lower stratosphere when injecting 5 Mt/yr of alumina particles with radius 240 nm (Figure 10b). For 25 Mt/yr of 240 nm particles or 5 Mt/yr of particles with 80 nm radius, these number concentrations reach values of up to 30 and 80 particles per $cm^3$, respectively (see Figure 10a and c, see Figure S11 in the supplement for corresponding resulting number densities from calcite injections). This is a substantial perturbation to the otherwise relatively clean air in the lower stratosphere and in the upper troposphere with background sulfuric acid aerosol number concentrations of about 10 per $cm^{-3}$ (Thomason and Peter, 2006) and ice nuclei concentration in the range of $10^{-1}$ to $10^{-4}$ per $cm^{-3}$ (DeMott et al., 2010). The injected particles will likely influence cirrus and polar stratospheric cloud abundances (e.g., Cziczo et al., 2019), an effect not accounted for by the model presented here. However, we account for heterogeneous chemistry on alumina and calcite surfaces (see Subsection "Heterogeneous Chemistry"). The total sulfuric acid SAD (i.e., sum of sulfuric acid coating and sulfuric acid aerosols) for injection rates of 5 Mt/yr and 25 Mt/yr alumina particles with 240 nm radius is not significantly different from the sulfuric acid aerosol SAD of the reference simulation (see Figure 10h, 10i and also Section 4.4). This is mostly due to the small angle $\beta$ for a constant contact angle ($\theta$) when the amount of sulfuric acid volume is small compared to the solid particle volume (i.e., for large alumina burden and large primary particle radius, see Figure 3). When emitting 5 Mt/yr of 80 nm particles, $\beta$ gets much larger and so does also the sulfuric acid surface area per particle (see Figure 10g). The alumina particle number density and SAD increase linearly with injection rate when keeping the radius constant. For the same injection rate, the number density is inversely proportional to the radius with a cubic power law, while the SAD increases linearly with decreasing particle radius, as observed in Figure 10a-f.

## 4.7  Ozone response to calcite and alumina particle injection

The resulting SAD presented in the previous section (see Figure 10) results in depletion of the total ozone column (TOC), which mainly correlates with the available alumina SAD (see Figure 11). Under present day ODS, injection of 5 Mt/yr of 80 nm particles and 25 Mt/yr of 240 nm particles both result in TOC depletion of more than 4% in the tropics and up to 16% and 12% in the polar regions, respectively. The baseline scenario, which injected 5 Mt/yr of alumina particles with radius of 240 nm only resulted in TOC depletion of less than 2% across all latitudes. Only the injection of 5 Mt/yr of alumina particles



**Figure 9.** The stratospheric sulfur cycle under conditions of SAI of 5 Mt/yr calcite particles with radii of 80 nm (blue), 160 nm (violet), 240 nm (red) and 320 nm (orange). All sulfur species (except $CaSO_4$) are shown in Gg Sulfur for burden (boxes) and Gg Sulfur per year for net fluxes (arrows). The solid species (colored boxes) are given in Gg of the corresponding material. The $HNO_3$ and HCl flux to $Ca(NO_3)_2$ and $CaCl_2$ are given in Gg $HNO_3$ per year and Gg HCl per year. Cross tropopause fluxes are calculated by balancing the mass balance of the individual species.

of 320 nm and 240 nm radius results in a smaller TOC depletion compared to the sulfur-based scenarios. The resulting RF from injection of alumina particles of this size is about 25%-33% smaller compared to the sulfur based scenarios at the same injection rates (see Figure 4b). For injection of 5Mt/yr 160 nm alumina particles the TOC depletion is only slightly enhanced compared to the sulfur based scenarios (Figure 11a), while resulting only in about 10% reduced RF compared to the sulfur based scenarios (see Figure 4b). When injecting 25 Mt/yr of alumina particles with radius 240 nm the ozone depletion is 50% larger compared to the injection of $SO_2$ (see Figure 11). However, these results are subject to large uncertainty (see Vattioni et al., 2023b) due to the lack of experimental data on heterogeneous chemistry on alumina particles.







**Figure 10.** The resulting zonal mean number densities (a-c), alumina SAD (d-f) and total sulfuric acid SAD (sum of sulfuric acid aerosols SAD and SAD from sulfuric acid coating on alumina particles, g-i) from injection of 5 Mt/yr of particles with 80 nm (a,d,f), 5 Mt/yr of particles with 240 nm particles (b,e,h) and 25 Mt/yr of particles with 240 nm radius (c,f,i). The same Figure for calcite particles is shown in the supplement in Figure S11.

All calcite injection scenarios result in an increase of TOC in the polar regions of up to about 6%, but almost no change at midlatitudes under present day ODS. This is mostly due to the removal of HCl from the stratosphere on calcite particles in agreement with the findings of Dai et al. (2020). However, the uptake of HCl, $HNO_3$ and $H_2SO_4$ (Reactions R4-R6) is the only heterogeneous chemistry process considered on calcite particles, which is a simplification. The resulting products will likely form hydrates (Cziczo et al., 2019), which may host other heterogeneous reactions such as R1-R3; our study only considers them on alumina particles. However, there is no experimental data on such reactions available for calcite surfaces, which makes the modelled response of the stratospheric ozone layer to calcite particle injections highly uncertain.



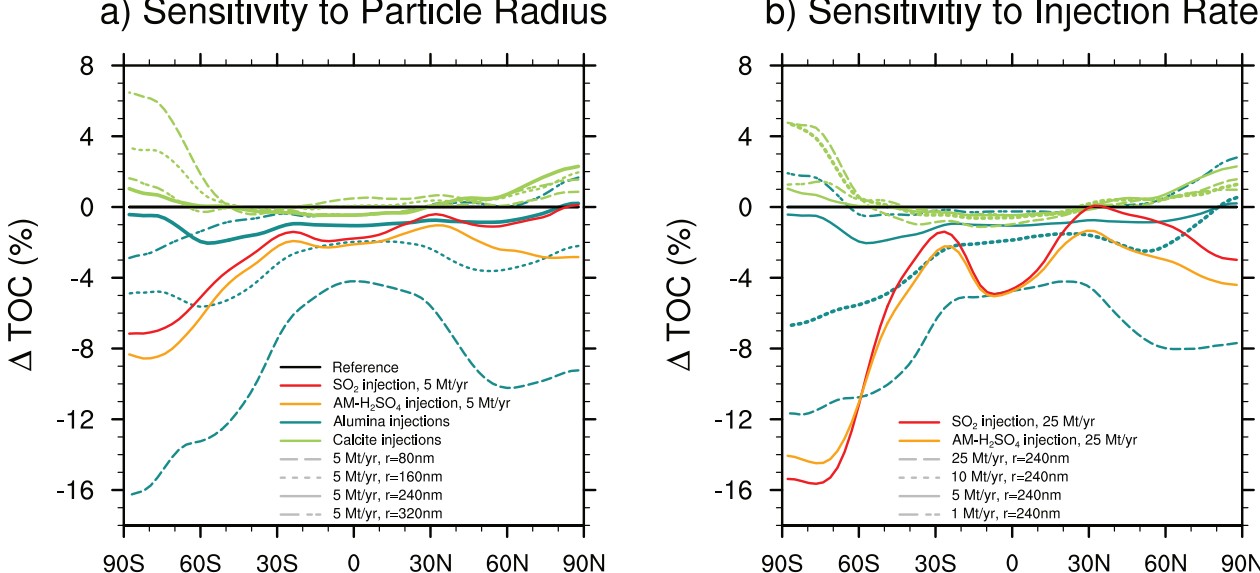

**Figure 11.** The simulated zonal mean total ozone column anomaly resulting from (a) injecting 5 Mt/yr of alumina and calcite particles with radii of 80 nm, 160 nm, 240 nm and 320 nm as well as 5 Mt/yr injections of $SO_2$ and $AM-H_2SO_4$ and (b) from injecting 1 Mt/yr, 5 Mt/yr, 10 Mt/yr and 25 Mt/yr calcite and alumina particles at radius of 240 nm as well as 25 Mt/yr $SO_2$ and $AM-H_2SO_4$ injections.

## 5 Conclusions

This study presents the first aerosol-chemistry climate-model incorporating an interactive solid particle microphysics scheme to investigate the risks and benefits of SAI of solid particles. The solid particles considered in this study are fully interactive with the stratospheric sulfur cycle. The model also allows for uptake of sulfuric acid at the particle surface via coagulation with

sulfuric acid aerosols and condensation of $H_2SO_{4(g)}$ on the particle surfaces, as well as for the formation of agglomerates via coagulation of solid particles. The solid particles are subject to advection, sedimentation and interactive wet and dry deposition in the troposphere. Furthermore, the model allows for representation of heterogeneous chemistry on the particle surface, and in particular, for the representation of the radiative effects of the particles, even after their microphysical interactions. The modular design of the model allows switching on and off the coupling of individual processes, which makes it perfectly suited

to investigate sensitivity and importance of the different processes relevant for the assessment of the risks and benefits of SAI of solid particles.

While this model was primarily developed for the evaluation of potential SAI scenarios of calcite and alumina injections, the model could also be adapted for representation of any other potential particle type or even for other applications. This could for example be the re-evaluation of radiative and chemical impacts of alumina particles emitted to the atmosphere from solid fuel

space shuttle rocket launches, which will likely increase significantly in the future (Jackman et al., 1998; Danilin et al., 2001;





Ross and Sheaffer, 2014), the evaluation of the growing impacts of microplastic nanoparticles transported in the atmosphere (Revell et al., 2021), analysis of wildfire impacts on stratospheric ozone (Solomon et al., 2023), and analysis of the role of meteoritic dust in the upper atmosphere (Biermann et al., 1996).

By using the model documented here, we show that the injection of solid particles likely results in significantly smaller stratospheric aerosol burden compared to the same injection rate of $SO_2$ and $AM-H_2SO_4$, even when injecting small particles with radius of 80 nm. This is mainly due to the larger average particle radius and the larger density of solid particles compared to sulfuric acid aerosols. Therefore, the corresponding net all sky ToA RF is largest for sulfur based injection scenarios when injecting the same amount of material per year (see Figure 4). Thus, alumina and calcite particles injected at a radius of 240 nm are only more effective in backscattering solar radiation per resulting aerosol burden, but not per injection rate of material.

Furthermore, we show that injection of solid particles to the stratosphere would deplete the stratospheric background sulfuric acid aerosol layer by more than 50% when injecting 5 Mt/yr of particles at 240 nm radius or smaller. Alumina particles would acquire a sulfuric acid coating through condensation of gaseous sulfuric acid on the particle surface and through coagulation of sulfuric acid aerosols with solid particles. The acquired sulfuric acid coating would have the equivalent thickness of about 10 nm if equally distributed over the resulting alumina SAD when injecting particles at 5 Mt/yr with radii of 240 nm. The

resulting coating thickness would be smaller when increasing the injection rate due to a larger alumina SAD to sulfuric acid ratio. However, a sulfuric acid coating distributed homogeneously over the alumina particles is unlikely due to a rather steep contact angle of about 30° of sulfuric acid on alumina surfaces (Vattioni et al., 2023b). Thus, it is likely that also some alumina surface would be available for heterogeneous chemistry.

Therefore, the response on TOC from alumina particle injections is largely dependent on the resulting alumina SAD, which

is a function of the alumina injection rate and the injected particle size. While for small injection rates a large fraction of the alumina particles would be covered by sulfuric acid, for large injection rates this fraction decreases significantly when assuming injection of alumina particles with radii of 240 nm. We assumed a realistic parameterization from (Vattioni et al., 2023b, dissociative, HCl only with $\alpha_{ClONO_2}$=0.1) for the heterogeneous reaction of $ClONO_2$ with HCl (reaction R1) on alumina SAD and the same heterogeneous chemistry on sulfuric acid coating as on sulfuric acid aerosols to quantify the expected TOC

alteration from alumina particle injections under present day ODS concentrations. Compared to the same injection rate of sulfuric acid aerosols, the resulting response of the the zonal mean TOC from injection of alumina particles is only smaller for small injection rates or large injected particle radii (see Figure 11).

For the injection of calcite particles we find similar perturbations to the stratospheric sulfur cycle as for alumina particles. However, the sulfuric acid taken up on calcite particles would react to $CaSO_4$. Assuming uptake coefficients of $10^{-4}$ for $HNO_3$

and $10^{-5}$ for HCl following Dai et al. (2020), 92% of average solid particle burden would remain in the form of $CaCO_3$ at injection rates of 5 Mt/yr $CaCO_3$ under present day ODS. This would likely not change the scattering properties of calcite particles, but could significantly alter heterogeneous chemistry hosted on the particle surface. Accounting for the uptake of HCl, $HNO_3$ and $H_2SO_4$ alone is not expected to alter stratospheric ozone significantly. However, heterogeneous chemistry on solid particles is not yet very well constrained due to the lack of experimental and observational data, which introduces

substantial uncertainty on the ozone response of solid particles.



The two biggest limitations of the model which result in major uncertainty of the presented results are the 1) missing interactions of the solid particles with clouds, such as polar stratospheric clouds and cirrus clouds as well as 2) the missing sub-grid scale microphysical injection plume-scale processes. Solid particles could serve as ice condensation nuclei for cirrus clouds in the upper troposphere after re-entry to the troposphere via sedimentation. Altering the cirrus cloud thickness could

result in a strong positive (cirrus cloud thickening) or a negative (cirrus cloud thinning) feedback on climate (Cziczo et al., 2019). Furthermore, the effect of solid particles on polar stratospheric clouds is unclear, but theoretically, the solid particles could also serve as cloud condensation nuclei for PSCs. It is only speculation whether this would result in overall less, but larger, or more, but smaller, PSCs. The latter case could for example result in less denitrification over the winter poles, which would result in less ozone depletion. This increases uncertainty of impacts on stratospheric ozone even more.

The second major limitation concerns the dispersion methods within the stratosphere (see also Blackstock et al., 2009). In contrast to a gas like $SO_2$, solid particles cannot be easily released to the stratosphere. They would require a carrier gas or a carrier liquid which could add further perturbation to stratospheric composition. Furthermore, the DEM modelling presented in this study (see Section S1 in the supplement) shows that it could be challenging to release solid particles to the stratosphere without immediate agglomeration. However, processes such as wind speed, turbulence, dilution and Van der Waals forces could

affect coagulation efficiencies. On the one hand, this could result in rapid formation of big agglomerates, which significantly reduce the stratospheric residence time as well as the backscattering efficiencies of the particles. On the other hand, particles could spend more time as monomers if collision speeds in the turbulent plume overcome the large Van der Waals forces of small particles. This limitation poses major uncertainty to the results presented here and they can only be addressed via injection plume modelling at the sub-grid scale or small scale field experiments such as proposed in Dykema et al. (2014).

With this study, we have shown that our model can be a useful tool to explore risks and benefits of SAI of solid particles. However, the results are still uncertain due to a number of limitations, such as lack of experimental data needed to refine the parameterizations of microphysical processes and heterogeneous chemistry. Given this model uncertainty, it is presently unclear whether SAI of alumina and calcite particles would result in smaller or larger side effects compared to sulfur-based SAI. This is in contrast to Arias et al. (2021, IPCC, AR6, WG1, Chapter 4, Page 629 ) which states: "Injection of non-sulphate

aerosols is likely to result in less stratospheric heating and ozone loss", but more in alignment with what was stated on SAI of solid particles in the latest ozone assessment report (WMO, 2022), which highlights the uncertainties. Given the potential benefits of solid particles over sulfuric acid aerosols we recommend conducting further research.



*Supplement.* The supplement related to this article is available online at https://doi.org/10.5194/amt-16-xxx-2023-supplement.

*Code and data availability.* The model code of SOCOL-AERv2 incorporating the presented solid particle microphysics scheme is available
in Vattioni et al. (2023a) and the simulation data presented in this study is available in Vattioni (2024).

*Author contributions.* SV wrote the paper draft, created most of the figures, developed the model, tested the model and did the data
analysis. RW performed some of the simulations, provided support with data analysis and created some of the figures. AF and AS
contributed to model code development, debugging and sanity checking. JAD performed the Mie-calculations to get the optical properties
for alumina and calcite particles and their agglomerates. BL helped with the implementation of microphysics such as the contact angle on
solid particles as well as implementation of optical properties. GAK and CAB run the DEM to determine the mobility radius of alumina and
calcite particles and all authors contributed to the discussion of the results.

*Competing interests.* At least one of the (co-)authors is a member of the editorial board of Geoscientific Model Development. Other than
this, the authors declare that they have no conflict of interest.

*Acknowledgements.* We especially thank Debra Weisenstein for discussions about her original solid particle AER code as well as David
Keith for valuable discussion of our results. We also thank Claudia Marcolli for bringing up the idea of representing the sulfuric acid
coating on alumina particles by accounting for the contact angle. Furthermore, we also thank David Verbart, Yaowei Li and Corey Pedersen
for discussions of the results. Support for Gabriel Chiodo and Andrea Stenke was provided by the Swiss Science Foundation within the
Ambizione grant no. PZ00P2_180043. Support for Sandro Vattioni was provided by the ETH Research grant no. ETH-1719-2 as well as by
the Harvard Geoengineering Research Program. JD was also supported by the Harvard Solar Geoengineerig Research Program. Timofei
Sukhodolov acknowledges the support from the Swiss National Science Foundation (grant no. 200020-182239) and the Karbacher Fonds,
Graubünden, Switzerland. GAK and CAB acknowledge the support from the Particle Technology Laboratory, ETH Zurich and, in part, the
Swiss National Science Foundation (grant no. 200020_182668, 250320_163243 and 206021_170729).



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
