# Peer review of "A fully coupled solid particle microphysics scheme for stratospheric aerosol injections within the aerosol-chemistry-climate-model SOCOL-AERv2"

_EGUsphere, 2024_

## Author Comment (AC1)

Dear Anton Laakso,

Thank you very much for your positive review and your very useful comments and suggestions, which have helped improve the document. We very much appreciate the time you invested. Please find below in blue detail answers to your comments in blue.

Sincerely,

Sandro Vattioni and Co-Authors

The manuscript by Vattioni et al. investigates the injection of solid particles, in particular alumina and calcite particles for stratospheric aerosol injection (SAI). They also implement a solid particle microphysics scheme on SOCOL-AER chemistry climate model, which is then coupled with the existing sulfur cycle, the heterogeneous chemistry scheme and radiative transfer model. This allows a relatively thorough and comprehensive review of solid particle injections. Alternative injection materials are an important issue for SAI and research into them is very welcome and needed. Despite the expected limitations and uncertainties in modelling the subject, this study is a major step forward in the field. Overall, this is an interesting, well written and really good study. It is therefore difficult to find anything to criticise or disagree with. Therefore, I have only few comments for authors to consider and recommend that this manuscript be published.

The only thing I would have liked to know a bit more about was stratospheric heating. As the authors state, this is one of the main interests of using alternative material instead of sulfur-based injections. If there are some restrictions on why stratospheric heating cannot be shown, it would be nice to see values for absorbed radiation. It is also interesting that based on Supplementary Figure S5 the absorption cross section for calcite is larger than for H2SO4 or alumina for the visible light spectrum. I am unable to estimate whether this leads to significantly greater absorption of shortwave radiation, but it would be interesting to see.

Thank you for this comment. We decided to not show plots on stratospheric warming, but to mainly focus on a detailed model description and evaluation. Since the reduced stratospheric warming is indeed one of the main benefits of solid particles that warrants detailed analysis, we decided that this merits a publication on its own, which allows for the required, detailed focus on this aspect. This follow up publication is currently in preparation and will be submitted soon.

The separate presentation of shortwave and longwave radiative forcing (e.g. in a supplement) could also be a good addition.

This will be highlighted, too, in the follow up publication.

Few comments on specific lines:

P2 L21-23: This may give the impression that these listed uncertainties are results that have been investigated and shown in this study.

We think it is important to list major uncertainties. We clearly state what the presented model is capable of doing with the next sentence: "The model presented in this work offers a useful tool for sensitivity studies and incorporating new experimental results on the SAI of solid particles"

P3 L56: Not only are they inefficient at backscattering solar radiation, but they can also absorb thermal radiation, which is not good.

The aspect of stratospheric heating is mentioned in the same listing under point 3. We added a side note that the absorption of LW also leads to reduced net radiative forcing: "… (3) absorption of

mainly outgoing terrestrial radiation reducing the net radiative forcing and resulting in stratospheric warming, which changes the large-scale atmospheric circulation and global and regional precipitation patterns (Aquila et al., 2014; Tilmes et al., 2017; Visioni et al., 2021; Jones et al., 2022; Wunderlin et al., 2024; Laakso et al., 2024) …"

P3 L62 Perhaps there could be some discussion in the conclusions as to whether solid aerosols offered a solution to these limitations based on this study. Of course, it is quite difficult to estimate e.g whether there are lower inter-model uncertainties in the case of solid materials than for sulfur injections based on this study, but there were some rather large differences between Weinstein et al. 2015 and this study that I found interesting.

We think most of these limitations (except inter-model differences and stratospheric heating) are discussed in the Conclusion section. See our answer to your comment on "P19 L480-490" and "P19 L504-506" on some explanation why we do not discuss inter-model differences to Weisenstein et al. 2015. If there were inter-model comparison studies on SAI of solid particles, these inter-model differences are likely much larger for SAI of solid particles compared to sulfur-based SAI since there are no observations for solid particles in the stratosphere. Many of the microphysical processes of solid particles are not known and characterized yet due to lack of observations and experimental work. Therefore, modelers need to come up with assumptions, which results in substantial inter-model uncertainty. This aspect will be highlighted in a follow up publication.

P3 L66: Just a comment: larger absorption leads to a reduction in (global) precipitation, which is especially important for larger injections (see e.g. Laakso, A et al: Earth Syst. Dynam., 15, 405-427, https://doi.org/10.5194/esd-15-405-2024, 2024). Personally, this is one of the main reasons why I think solid materials could be an interesting alternative to sulfur.

Thanks for pointing to this. We restructured the paragraph slightly so that the reduced stratospheric warming is highlighted as the major benefit from solid particle injections: "Most importantly, the absorption efficiency of longwave (LW) and shortwave (SW) radiation per resulting aerosol burden is significantly smaller for many solid materials compared to sulfuric acid aerosols, resulting in reduced stratospheric warming.»

We added your suggested publication to the reference list, when first mentioning the impacts of stratospheric warming, and specifically highlighted the global precipitation aspect (page 3, line 58 to 60).

P6 L169: Emissions of SW radiation?

Thanks for spotting this. We corrected this typo. The emission accounts only for the LW code.

P7 L201: Is the lack of aerosols in the largest bins mainly due to lack of agglomeration? "Large number"-mers are quite large if the monomer radius is 200nm and I assume their lifetime in the atmosphere is not long.

It is mainly due to reduced coagulation. The stratospheric residence time of the aerosols is still about 0.8 years for alumina and 1 year for calcite when injecting 5 Mt/yr of 240 nm particles. But Figure 3 clearly shows that there is only little agglomeration. The large particles are relatively immobile and do not coagulate significantly with each other. But you are right, if there was more efficient coagulation, this would also reduce the stratospheric residence time of the particles.

P12 L315-323: It may just be me, but I had some problems understanding how the alumina particles are presented, but I have no suggestion as to how this could be made clearer. It may be that I was

confused by the monomers and agglomerates and thought at first that they were something else than different bins in the same 10-bin distribution. It was also only later that I realized that the injections were indeed always made in the first bin, regardless of the size of the particles. However, unless I am completely wrong, there are now 10 bins for Alumina (partially coated particles), H2SO4 (partially coated), Alumina (fully coated), H2SO4 (fully coated)? A figure would always be helpful, but I am not sure if it is that important here. Maybe for a supplement at most.

You got everything right. We don't think an additional figure is needed here. The paragraph in the beginning of Section 2.2 (page 7, line 200-204) should make it clear. We clearly state: "Particles are always injected as monomers, which can grow to larger order agglomerates via coagulation (see subsections on "Coagulation"). The injected monomer radius can be specified in the model via a namelist parameter and varies between 80 nm and 320 nm in this study to investigate trade offs between agglomeration, sedimentation speed and backscatter efficiency of different injected monomer radii. To keep track of the monomers and their agglomerates the solid particles are represented by different mass bins (i=1-10), with mass doubling between subsequent bins (i.e., 1-, 2-, 4-, 8-, 16-, 32-, 64-, 128-, 256- and 512-mers)."
However, for clarity we have added references to "mass bin 1" for monomers and "mass bin 2 to 10" for the agglomerates as well as to "Section 2.2" in Section 2.3, first paragraph between (page 12, line 322 to 330).

P13-14 L357-361. When I read this, I wondered where the contact angle came from. Later in the results section it is found that it is predefined. It could be mentioned here.

Thank you very much for pointing to this. The contact angle of $H_2SO_4$ on alumina was measured and found to be about 30° in Vattioni et al. (2023). We added this information with the sentence: "Vattioni et al. 2023b measured the contact angle ($\theta$) of $H_2SO_4$–$H_2O$ on alumina surfaces as a function of weight percent and found $\theta$ to be 31°±7°."

P17 L445 CasO_4 -> CaSO_4

Thanks for spotting.

P19 L480-490 Since the model has been updated, it is not surprising that the burdens are different (lower) than in Weisenstein et al. (2015). However, if I have calculated and understood the units correctly (there is a good chance that I haven't), the sulfur burdens in your study are slightly higher than in Weisenstein et al. (2015). But this is just interesting (if true) and I don't mean that you should open up differences in every single process between models to explain it. But if you had an idea for a reason, you could mention it.

The aerosol-microphysics module, AER, of SOCOL-AER has undergone extensive further development since its implementation into SOCOL-AER, while the study of Weisenstein et al., 2015 is still based on the original AER model (Weisenstein et al., 1997 and 2007). There were already several updates made to the AER code upon implementation into SOCOL-AER, which are described in detail in Sheng et al. 2015. For example, the simple upwind method used in the original AER code to describe sedimentation was replaced by the numerical scheme of Walcek et. al., 2000 to reduce numerical diffusion. SOCOL-AER results in almost identical sedimentation velocities for sulfuric acid aerosols as well as alumina particles as shown in Figure 4 from Weisenstein et al. 2015 (see our supplementary Figure S4 for comparison). However, this figure is based on the simple equation of the termination setting velocity, which is applied in the models, and which is not difficult to reproduce. Differences in sedimentation flux mainly result from how the sedimentation scheme is implemented into the model (mass transport from level to level with decreasing level heights towards the ground).

But the stratospheric aerosol burden is not dependent on sedimentation rates only, but it is also the dynamics and thus transport, which significantly plays in. In contrast to SOCOL-AER, the chemistry transport model 2D-AER is only two dimensional (i.e., zonally averaged) and thus, has prescribed dynamics and does not account for dynamical impacts from stratospheric warming, which is especially important in the case of sulfur-based SAI.

Also, the sulfuric acid coagulation scheme has received several updates upon implementation into SOCOL-AERv1. In SOCOL-AER coagulation is solved by a semi-implicit method (Jacobson and Seinfeld, 2004), which is different from the original 2D-AER code. Later, Feinberg et al. 2019 developed the model further and improved mass conservation in the sedimentation scheme. Additionally, in the new version of SOCOL-AER (version2, used in our study) Feinberg et al. 2019 accounted for interactive wet and dry deposition, whereas the previous version and the 2D-AER model removed sulfuric acid aerosols and solid particles by assuming a prescribed aerosol lifetime in the troposphere. Taking into account interactive dry and wet deposition also significantly changes the resulting stratospheric aerosol burden for unperturbed conditions (Feinberg et al. 2019). The interactive wet and dry deposition schemes probably are less important for simulation of volcanic eruptions and stratospheric solar radiation modification, but that has not been investigated.

When we looked into the 2D-AER code we also found that coagulation between larger fractals of "coated" alumina particles with smaller fractals of "pure" alumina particles was not accounted for in Weisenstein et al. 2015. We do not know to what extent this has an influence on the resulting size distributions in 2D-AER. We think that other aspects such as different representation of the fractal dimension as well as different calculation of coagulation kernel between the two model versions only play a minor role when explaining differences between the two models since this would not result in first order effects.

In SOCOL-AERv2, there are many more updates concerning sulfur chemistry (aqueous sulfur chemistry in clouds, updated photolysis and reaction rates of sulfur compounds and other species) and sulfuric acid microphysics (composition of individual aerosol particles allowed to be size-dependent, taking into account wet aerosol radii >3.2 mm for sedimentation, improvement of representation of the super-cooled liquid aerosol fraction, change from dry binning to wet binning, fixing bugs in the condensation scheme) which do not affect results from SAI by solid particles but likely the results from sulfur-based SAI simulations in comparison with 2D-AER. The sulfuric acid aerosol size distribution is also affected by the zonal mean representation in 2D-AER, which does not provide the natural variability in particle sizes (among others due to latitudinal T and RH differences). This leads to generally higher sulfur sedimentation fluxes in the 3D work given the highly nonlinear dependence of sedimentation flux on particle size (J proportional to $r^5$). Finally, this recent publication highlights how the call sequence and the microphysical time step can influence the resulting aerosol burden and size distribution as well as RF from sulfur-based SAI (Vattioni et al., 2024). The suggested improvements were already accounted for in this study, but not in 2D-AER.

Long story short, even though the microphysics for both, solid particles and sulfuric acid aerosols in SOCOL-AERv2 originated from the 2D-AER model, these schemes have undergone multiple updates and further developments which make it very difficult to compare result from the two models (see detailed description of comparisons in Sheng et al., 2015 and Feinberg et al., 2019). The main differences probably result from the different dynamical core between the two models (ECHAM5.4 in SOCOL-AERv2 and prescribed zonal mean transport and 2D eddy diffusivities in 2D-AER), which result in different transport (e.g., different upwelling velocities). Furthermore, we use year 2020

boundary conditions for GHG, SST, SIC and ODS, while Weisenstein et al. 2020 used prescribed year 2000 boundary conditions for dynamics.

[Figure]

The figure above (i.e., Figure 5a in Weisenstein et al. 2015) shows the stratospheric aerosol burden resulting from the 2D-AER for various injection rates. However, burden is shown in Tg S for the $SO_2$ injection scenarios and for the $H_2SO_4$ injection scenario from Pierce et al., 2010 with corresponding injection rates in Tg S/yr in the form of $SO_2$ and $H_2SO_4$, whereas in our manuscript we show injections in Tg $SO_2$/yr and Tg $H_2SO_4$/yr as well as total resulting $H_2SO_4$-$H_2O$ burden. Therefore, to allow for comparison with our study, both the burden and the injection rate must be converted to absolute/total masses of $SO_2$ and $H_2SO4$-$H_2O$. We have linearly extrapolated the violet and yellow line in the Figure 5a from Weisenstein et al. 2015 to burden in Tg $H_2SO_4$ (not accounting for $H_2O$) and injection in Tg $H_2SO_4$/yr (thick yellow line) and Tg $SO_2$/yr (thick violet line). In Weisenstein et al. 2015 the sulfur-based scenarios result in similar burden as the alumina injection scenario with r=160 nm, even though sulfuric acid has a much lower density (1.8 g/cm$^3$) compared to alumina (3.95 g/cm$^3$) and thus should settle more slowly if the radius was the same. Weisenstein et al. 2015 write "Alumina monomers fall at a faster rate than sulfate particles of the same diameter, given their greater density (3.8 g cm$^{-3}$ for Al2O3, approximately 1.7 g cm$^{-3}$ for stratospheric $H_2SO_4$–$H_2O$ particles)". However, implications for stratospheric aerosol residence time and burden are not further discussed. For comparison we also added dots for the results from SOCOL-AERv2 into the figure above (also not accounting for $H_2O$ mass within $H_2SO_4$-$H_2O$). Our sulfur-based scenarios result in similar burden as the sulfur-based scenarios in Weisenstein et al., 2015. However, our alumina-based scenarios result in lower burden, which appears to be more physical.

P19 L504-506 It is also interesting to note that in this study, the injection of 80nm particles resulted in a much larger radiative forcing than in Weisenstein et al 2015. Do you know why this is? The burden was larger in Weisenstein et al., but on the other hand, in this study the mass fraction is largest in 16mers, whereas in Weisenstein et al. it was 64mers (with 4 Tg/yr). This is probably the answer. If so, this is another nice example of how important aerosol microphysics is, even when simulating solid particles.

[Figure]

[Figure]

This might be an explanation. However, given the substantial differences between the models (see also previous comment), it would be pure speculation to conclude that the degree of agglomeration was the first order factor for differences in RF. It is important to note that in Weisenstein et al., 2015 the radiative transfer calculations were performed offline, based on the resulting spatial aerosol size distribution from the 2D-AER model. They also only show the top of the atmosphere clear sky short wave RF, whereas we show the net top of the atmosphere all sky RF. Weisenstein et al. 2015 use the same Mie-scattering code for calculation of optical properties (Rannou et al., 1999) and the same LW radiative transfer code (Mlawer et al., 1997) for calculation of stratospheric heating as is used in SOCOL-AERv2. However, the SW radiative transfer code they used (Charlson et al., 1991) is different from the one in SOCOL-AERv2, rendering comparisons of RF values difficult. Above, we are copy-pasting Figure 1a (left) and 6a (right) from Weisenstein et al. (2022) which show differences in burden (left) and resulting all sky RF (right) from the very same sulfur-based emission scenarios performed by three different aerosol-chemistry climate models. The results between ECHAM and SOCOL differ by up to a factor 1.3-1.4 in burden and a factor 1.8-2.0 in RF, even though ECHAM and SOCOL use the same radiative transfer code. Compared to CESM (with different radiative transfer code) differences are even larger (factor 2 for burden and 4 for RF). Furthermore, all three models apply different aerosol-microphysics modules. Therefore, we are not surprised to see such differences also when comparing 2D-AER with SOCOL-AERv2. Given the countless potential sources of model differences we decided to not further comment about potential reasons.

P21 L526. These fractions of depleted sulfate burdens might also be worth mentioning in calcite simulations, although they can also be seen in Fig. 9. It is also interesting that the fraction of depleted H2SO4 mass is larger in alumina than in calcite for 80 nm injections, but vice versa for e.g. 320 nm injections.

We added this sentence to Section 4.5: "This results in depletion of the background stratospheric sulfuric acid aerosol layer of 90%, 72%, 53%, 38% for injection of particles with radius of 80 nm, 160 nm 240 nm and 320 nm, respectively (see Figure 9)."

P26 L610 TOC already used in P25 L560, so could be opened there

We now introduce the abbreviation "TOC" upon the first mention and use it thereafter.

P31 L681 "…on polar stratospheric clouds (PSC) is unclear.." <--please add "(PSC)"

We now introduce the abbreviation "PSC" on Page 4, line 115 and use it thereafter.

**References:**

Charlson, R. J., Langner, J., Rodhe, H., Leovy, C. B., and Warren, S. G.: Perturbation of the northern hemisphere radiative balance by backscattering from anthropogenic sulfate aerosols, Tellus A, 43, 152–163, 1991.

Feinberg, A., T. Sukhodolov, B. P. Luo, E. Rozanov, L. H. E. Winkel, T. Peter, and A. Stenke (2019): Improved tropospheric and stratospheric sulfur cycle in the aerosol-chemistry-climate model SOCOL-AERv2, Geosci. Model Dev., DOI:10.5194/gmd-2019-138.

Rannou, P., McKay, C. P., Botet, R., and Cabane, M.: Semiempirical model of absorption and scattering by isotropic fractal aggregates of spheres, Planet. Space Sci., 47, 385–396, 1999

Mlawer, E. J., Taubman, S. J., Brown, P. D., Iacono, M. J., & Clough, S. A. (1997). Radiative transfer for inhomogeneous atmospheres: Rrtm, a validated correlated-k model for the longwave. J. Geophys. Res., 102 , 6,663–16,682. doi: 10.1029/97JD00237

Sheng, J., D. Weisenstein, B. Luo, E. Rozanov, A. Stenke, J. Anet, H. Bingemer, and T. Peter (2015): Global atmospheric sulfur budget under volcanically quiescent conditions: Aerosol-chemistry-climate model predictions and validation (2015), J. Geophys. Res., DOI:10.1002/2014JD021985.

Vattioni, S., Luo, B., Feinberg, A., Stenke, A., Vockenhuber, C., Weber,R., et al. (2023). Chemical impact of stratospheric alumina particle injection for solar radiation modification and related uncertainties. Geophysical Research Letters, 50, e2023GL105889. https://doi.org/10.1029/2023GL105889

Vattioni, S., Stenke, A., Luo, B., Chiodo, G., Sukhodolov, T., Wunderlin, E., and Peter, T.: Importance of microphysical settings for climate forcing by stratospheric $SO_2$ injections as modeled by SOCOL-AERv2, Geosci. Model Dev., 17, 4181–4197, https://doi.org/10.5194/gmd-17-4181-2024, 2024.

Walcek, C. J.: Minor flux adjustment near mixing ratio extremes for simplified yet highly accurate monotonic calculation of tracer advection, J. Geophys. Res., 105, 9335–9348, https://doi.org/10.1029/1999JD901142, 2000.

Weisenstein, D. K., Keith, D. W., & Dykema, J. A. (2015). Solar geoengineering using solid aerosol in the stratosphere. Atmospheric Chemistry and Physics, 15(20), 11835–11859. https://doi.org/10.5194/acp-15- 11835-2015

Weisenstein, D. K., Visioni, D., Franke, H., Niemeier, U., Vattioni, S., Chiodo, G., Peter, T., and Keith, D. W.: An interactive stratospheric aerosol model intercomparison of solar geoengineering by stratospheric injection of SO2 or accumulation-mode sulfuric acid aerosols, Atmos. Chem. Phys., 22, 2955–2973, https://doi.org/10.5194/acp-22-2955-2022, 2022.

---

## Author Comment (AC2)

Dear reviewer,

Thank you very much for your review and positive feedback on the manuscript and the comments that helped to improve it. We appreciate the time you took to review the manuscript. Below is our response to your review in blue.

However, while the paper is comprehensive, it would be useful to include more information on the validation of the model and comparison with observational data. I believe that by adding information on how well the model can reproduce the observed data, the reliability of both the model and the paper will be enhanced, so please consider this aspect.

We appreciate your comment and your suggestion. Unfortunately, there are no observations for solid particle injections into the stratosphere. Thus, it is not possible to compare the model to observations. However, potential limitations and uncertainties resulting from agglomeration in sub-ESM plume processes after injection of solid particles e.g. from an aircraft are discussed in detail in the "Discussion"-Section of the manuscript. Concerning the validation of the model there are many previous papers which demonstrate for example that SOCOL is accurately representing present-day climate (Stenke et al., 2013, Sukhodolov et al., 2021), stratospheric chemistry and ozone (Friedel et al., 2022), stratospheric aerosol burden and size distributions (Brodowsky et al., 2024) and deposition (Feinberg et al., 2019) as well as the effects of volcanic eruptions (Sukhodolov 2018, Clyne et al. 2021, Quaglia et al., 2022). The solid particle model presented here was thoroughly sanity checked against the original version SOCOL-AERv2 (Feinberg et al., 2019) by performing simulations with the same initial and boundary conditions. Given the many previous publications which demonstrate the performance of SOCOL-AER as well as the detailed discussion of limitations of the model in the last section we think that the model is sufficiently validated. We added a sentence to the manuscript in Section 2 (first paragraph on page 5, line 141 to 145) to point to these validation papers:

"Despite the lack of in-situ solid particle measurements in the stratosphere to evaluate the solid particle module, the SOCOL models have been extensively evaluated against observations for climate (Stenke et al., 2013, Sukhodolov et al., 2021, Morgenstern et al., 2022), stratospheric chemistry (Friedel et al., 2022), background aerosol (Brodowsky et al., 2024) and volcanic aerosol (Sukhodolov 2018, Clyne et al., 2021, Quaglia et al., 2022) in the past."

Sincerely,

Sandro Vattioni and Co-Authors

**References:**

Brodowsky, C. V., Sukhodolov, T., Chiodo, G., Aquila, V., Bekki, S., Dhomse, S. S., Höpfner, M., Laakso, A., Mann, G. W., Niemeier, U., Pitari, G., Quaglia, I., Rozanov, E., Schmidt, A., Sekiya, T., Tilmes, S., Timmreck, C., Vattioni, S., Visioni, D., Yu, P., Zhu, Y., and Peter, T.: Analysis of the global atmospheric background sulfur budget in a multi-model framework, Atmos. Chem. Phys., 24, 5513–5548, https://doi.org/10.5194/acp-24-5513-2024, 2024.

Clyne, M., Lamarque, J.-F., Mills, M. J., Khodri, M., Ball, W., Bekki, S., Dhomse, S. S., Lebas, N., Mann, G., Marshall, L., Niemeier, U., Poulain, V., Robock, A., Rozanov, E., Schmidt, A., Stenke, A.,

Sukhodolov, T., Timmreck, C., Toohey, M., Tummon, F., Zanchettin, D., Zhu, Y., and Toon, O. B.: Model physics and chemistry causing intermodel disagreement within the VolMIP-Tambora Interactive Stratospheric Aerosol ensemble, Atmos. Chem. Phys., 21, 3317–3343, https://doi.org/10.5194/acp-21-3317-2021, 2021.

Feinberg, A., Sukhodolov, T., Luo, B.-P., Rozanov, E., Winkel, L. H. E., Peter, T., and Stenke, A.: Improved tropospheric and stratospheric sulfur cycle in the aerosol–chemistry–climate model SOCOL-AERv2, Geosci. Model Dev., 12, 3863–3887, https://doi.org/10.5194/gmd-12-3863-2019, 2019.

Friedel, Marina, Gabriel Chiodo, Andrea Stenke, Daniela I. V. Domeisen, Stephan Fueglistaler, Julien G. Anet, Thomas Pete "Springtime arctic ozone depletion forces northern hemisphere climate anomalies. Nature Geoscience 15.7, 541-547, 2022

Morgenstern, O., Kinnison, D. E., Mills, M., Michou, M., Horowitz, L. W., Lin, P., et al. (2022). Comparison of Arctic and Antarctic stratospheric climates in chemistry versus no-chemistry climate models. Journal of Geophysical Research: Atmospheres, 127, e2022JD037123. https://doi.org/10.1029/2022JD037123

Quaglia, I., Timmreck, C., Niemeier, U., Visioni, D., Pitari, G., Brodowsky, C., Brühl, C., Dhomse, S. S., Franke, H., Laakso, A., Mann, G. W., Rozanov, E., and Sukhodolov, T.: Interactive stratospheric aerosol models' response to different amounts and altitudes of $SO_2$ injection during the 1991 Pinatubo eruption, Atmos. Chem. Phys., 23, 921–948, https://doi.org/10.5194/acp-23-921-2023, 2023

Stenke, A., Schraner, M., Rozanov, E., Egorova, T., Luo, B., and Peter, T.: The SOCOL version 3.0 chemistry–climate model: description, evaluation, and implications from an advanced transport algorithm, Geosci. Model Dev., 6, 1407–1427, https://doi.org/10.5194/gmd-6-1407-2013, 2013.

Sukhodolov, T., Sheng, J.-X., Feinberg, A., Luo, B.-P., Peter, T., Revell, L., Stenke, A., Weisenstein, D. K., and Rozanov, E.: Stratospheric aerosol evolution after Pinatubo simulated with a coupled size-resolved aerosol–chemistry–climate model, SOCOL-AERv1.0, Geosci. Model Dev., 11, 2633–2647, https://doi.org/10.5194/gmd-11-2633-2018, 2018.

Sukhodolov, T., Egorova, T., Stenke, A., Ball, W. T., Brodowsky, C., Chiodo, G., Feinberg, A., Friedel, M., Karagodin-Doyennel, A., Peter, T., Sedlacek, J., Vattioni, S., and Rozanov, E.: Atmosphere–ocean–aerosol–chemistry–climate model SOCOLv4.0: description and evaluation, Geosci. Model Dev., 14, 5525–5560, https://doi.org/10.5194/gmd-14-5525-2021, 2021.